# R-Stitch: Dynamic Trajectory Stitching for Efficient Reasoning

## Abstract

Chain-of-thought (CoT) enhances the problem-solving ability of large language models (LLMs) but incurs substantial inference cost due to long autoregressive trajectories. Existing acceleration strategies either shorten traces via early stopping or compression, or adopt speculative decoding with a smaller model. However, speculative decoding provides limited gains when model agreement is low and rigidly enforces token-level consistency, overlooking the observation that some smaller models, when correct, produce significantly more concise reasoning traces that could reduce inference length. We introduce *R-Stitch*, a training-free hybrid decoding framework that leverages token-level entropy as an uncertainty proxy to delegate computation between a small language model (SLM) and an LLM. Our analysis shows that high-entropy tokens are more likely to induce errors, motivating an entropy-guided routing strategy that lets the SLM efficiently handle low-entropy tokens while delegating uncertain ones to the LLM, thereby avoiding full rollbacks and preserving answer quality. We further extend this design with *R-Stitch$^+$*, which learns an adaptive routing policy to adjust the token budget dynamically beyond fixed thresholds. By jointly reducing per-token decoding complexity and the number of generated tokens, our method achieves substantial acceleration with negligible accuracy loss. Concretely, it attains peak speedups of $3.00\times$ on DeepSeek-R1-Distill-Qwen-7B, $3.85\times$ on 14B, and $4.10\times$ on QWQ-32B while maintaining accuracy comparable to full LLM decoding. Moreover, it naturally enables adaptive efficiency–accuracy trade-offs that can be tailored to diverse computational budgets without retraining.

## 1 Introduction

Large language models (LLMs) have achieved impressive performance on a wide range of reasoning tasks, particularly when combined with chain-of-thought (CoT) prompting (Wei et al., 2022; Zhang et al., 2024; Zheng et al., 2024). By generating intermediate reasoning steps token-by-token, CoT enables LLMs to solve complex problems in arithmetic, logic, and code generation (Wei et al., 2022; Kojima et al., 2022; Zhang et al., 2023). However, this autoregressive decoding process is inherently slow, as each token requires a full forward pass through the model (Liu et al., 2024; Sadhukhan et al., 2024; Chen et al., 2023). The latency becomes especially problematic in CoT, where outputs often span thousands of tokens, significantly limiting the applicability of LLMs in time-sensitive scenarios.

To address this bottleneck, recent efforts have primarily focused on three directions: reducing the number of generated tokens, applying speculative decoding to accelerate the generation process, and optimizing KV cache access during long-context decoding. The first direction includes approaches that aim to shorten CoT sequences, such as early exiting or incorporating length-aware reward functions during Reinforcement Learning (RL) (Fatemi et al., 2025; Yang et al., 2025b; Ma et al., 2025; Yi et al., 2025; Jiang et al., 2025). The second leverages a small language model (SLM) to draft multiple tokens ahead, which are then verified in parallel by a larger LLM (Liu et al., 2024; Sadhukhan et al., 2024; Chen et al., 2023). If verification succeeds, the drafts are accepted; otherwise, generation rolls back to the last matching token. The third direction addresses the I/O bottleneck introduced by repeated access to large key-value (KV) caches during decoding, which becomes increasingly expensive in long-context reasoning. Techniques such as sparse or selective KV caching (Gao et al., 2025; Tang et al., 2024) have been proposed to reduce memory traffic and improve decoding speed on modern hardware.

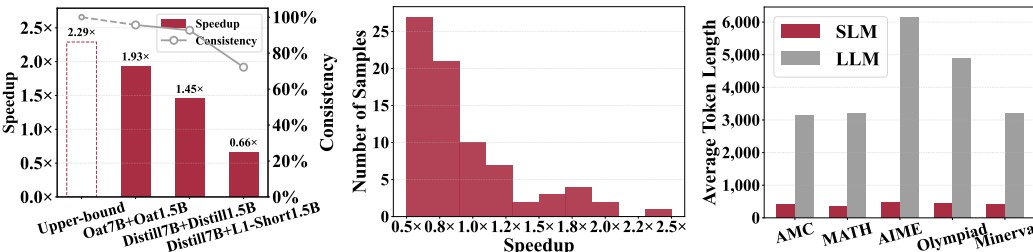

(a) Token-level consistency versus speedup across different LLMs.

(b) Speedup across individual samples in AMC.

(c) Token usage on questions answered correctly by SLM and LLM.

Figure 1: **Token-level consistency and speedup analysis.** (a) shows the relationship between token-level consistency and speedup in speculative decoding across different LLM-SLM pairs on AMC. (b) presents the distribution of speedup ratios across individual samples from AMC. (c) illustrates the token counts for questions correctly answered by both the SLM and LLM.

Among the three directions, speculative decoding has received considerable attention due to its potential for substantial speedups. However, its effectiveness critically depends on the consistency between the small language model (SLM) and the large language model (LLM). We quantify this limitation using token-level consistency, defined as the percentage of tokens for which the SLM produces the same output as the LLM given an identical prefix. Figure 1a presents token-level consistency and decoding speedup in speculative decoding across four model combinations on the AMC dataset: DeepSeek-R1-Distill-Qwen-1.5B/7B (Guo et al., 2025), L1-1.5B-Short (Aggarwal & Welleck, 2025) and Qwen2.5-Math-1.5B/7B-Oat-Zero (Liu et al., 2025). The results show a clear trend: lower consistency correlates with smaller speedup, particularly in reasoning-intensive tasks. Figure 1b shows the distribution of speedup ratios across individual samples from the AMC dataset, using DeepSeek-R1-Distill-Qwen-7B as the LLM and L1-Short as the SLM. While some samples achieve noticeable acceleration, a large number have speedups below $1\times$, indicating that speculative decoding can introduce overhead on certain inputs. Furthermore, Figure 1c compares the number of tokens generated by the same SLM and LLM on questions both models answer correctly. The SLM produces much shorter completions, suggesting that speculative decoding's rigid reliance on exact token agreement may prevent it from utilizing the SLM's more concise reasoning effectively.

To more flexibly exploit SLM for acceleration, we propose R-Stitch, an entropy-guided decoding framework inspired by the preceding observations. Our empirical analysis shows that tokens with higher entropy are more error-prone, which motivates a token-level routing strategy: the SLM acts as the primary generator, and the LLM is invoked only when needed. In this framework, confident low-entropy tokens are accepted directly from the SLM, while uncertain high-entropy tokens trigger LLM intervention for correction and continued generation. Compared with agreement-based speculative decoding, this dynamic delegation avoids full-sequence rollbacks, preserves the complementary strengths of both models, and achieves efficient reasoning under tight computational budgets.

To refine the entropy-based heuristic, we introduce R-Stitch$^+$. Rather than relying on a fixed threshold, R-Stitch$^+$ equips high-uncertainty tokens with a lightweight router that determines whether LLM intervention is necessary. The router is trained via RL with a latency-aware reward, enabling a data-driven policy that adaptively balances accuracy and efficiency. Thus, R-Stitch$^+$ extends the heuristic rule of R-Stitch into a more effective routing mechanism under diverse conditions.

We validate our proposed method on five challenging mathematical reasoning benchmarks using DeepSeek-R1-Distill-Qwen models at 7B, 14B, and 32B scales. Across all settings, our approach achieves substantial latency reduction while maintaining accuracy close to full LLM decoding, reaching peak speedups of $3.00\times$, $3.85\times$, and $4.10\times$ on the 7B, 14B, and 32B models, respectively. Overall, these results confirm that token-level, entropy-guided collaboration provides a principled and effective solution for accelerating CoT reasoning, addressing the limitations of speculative decoding and enabling flexible deployment under different computational budgets.

Our contributions are summarized as follows:

- We analyze the limitations of speculative decoding in low-consistency CoT settings and show that its rigid alignment with the LLM can sacrifice potential efficiency gains from the SLM.

- We analyze the entropy distribution of tokens in CoT reasoning and reveal its strong correlation with prediction errors, motivating entropy as an effective routing signal.

- We propose R-Stitch, an entropy-guided hybrid decoding paradigm that adaptively switches between the SLM and LLM to accelerate CoT generation without requiring additional training. We further extend it to R-Stitch$^+$, which learns a routing policy with a latency-aware RL reward, enabling more optimal efficiency–accuracy trade-offs.

- Extensive experiments on mathematical reasoning benchmarks demonstrate that our method achieves consistently lower latency at the same level of accuracy compared with speculative decoding.

## 2 RELATED WORK

**LLM reasoning.** Recent advances in prompting strategies have significantly improved the reasoning capabilities of LLMs, enabling them to solve complex tasks through structured intermediate computation. A variety of inference-time strategies have been proposed to enhance reasoning, primarily including Chain-of-Thought (CoT) (Wei et al., 2022; Zhang et al., 2024; Zheng et al., 2024), Tree-of-Thought (Yao et al., 2023), and Monte Carlo Tree Search (MCTS)-based decoding (Zhang et al.). Among these, CoT has emerged as a widely adopted method that guides the model to verbalize intermediate steps instead of directly predicting the final answer (Wei et al., 2022; Kojima et al., 2022; Zhang et al., 2023; Li et al., 2025a), improving performance on arithmetic and code reasoning benchmarks. However, these approaches often rely on token-by-token autoregressive decoding of long reasoning chains, which can be inefficient and lead to high inference latency, limiting their scalability in real-world applications.

**Accelerating reasoning in LLMs.** Improving the efficiency of chain-of-thought (CoT) reasoning involves addressing three key bottlenecks: reasoning length, per-token decoding cost, and KV cache I/O overhead. The first focuses on reducing the number of generated tokens by shortening reasoning chains. Methods such as Concise Reasoning (Fatemi et al., 2025) and ShorterBetter (Yi et al., 2025) use length-aware rewards to promote brevity without sacrificing correctness. Other approaches, including DEER (Yang et al., 2025b) and ThinkPrune (Hou et al., 2025), leverage model entropy or attention patterns to eliminate unnecessary steps, while methods like AdaptThink (Zhang et al., 2025) and ThinkLess (Fang et al., 2025) aim to skip reasoning entirely when it is not needed. These techniques effectively reduce sequence length, thereby lowering total decoding time. The second line of work targets per-token decoding complexity. Speculative decoding (Liu et al., 2024; Sadhukhan et al., 2024; Chen et al., 2023) speeds up generation by having a small language model (SLM) propose token drafts, which are verified in parallel by a large LLM. However, low agreement between the SLM and LLM often causes frequent rollbacks, limiting acceleration gains. Finally, recent research (Gao et al., 2025) (Tang et al., 2024)has highlighted the importance of reducing KV cache I/O—an increasingly dominant cost in long-context decoding, which also happens during the inference phase of reasoning models due to the length of CoT. Since decoding is often I/O-bound, reducing the loading of KV cache from global memory to on-chip memory of GPUs (e.g., via sparse or selective KV caching) can significantly improve system throughput and the speed of generation.

**Stitching.** Stitching broadly refers to composing multiple models or modules into a unified inference process, often by connecting their intermediate representations or generation trajectories. Early approaches like SN-Net (Pan et al., 2023), SN-Net v2 (Pan et al., 2024), and ESTA (He et al., 2024b) focused on layer-wise stitching within single-modality models, enabling efficient adaptation across backbone variants. Other works such as T-Stitch (Pan et al., 2025b) extended stitching to the trajectory level, applying diffusion-based denoising to merge reasoning paths. Beyond single domains, cross-model stitching has also emerged—LLaVA (Liu et al., 2023), for example, connects a vision encoder with an LLM to build a multi-modal agent. More recent efforts explore general-purpose stitching frameworks, such as MetaQuery (Pan et al., 2025a) and UniWorld (Lin et al., 2025), which introduce connector modules (e.g., MLPs or learned queries) to link heterogeneous models. These advances highlight stitching as a general paradigm for unifying diverse model components. Our work follows this trend by stitching a small and large language model at the token level during autoregressive decoding, enabling efficient collaboration without expensive retraining.

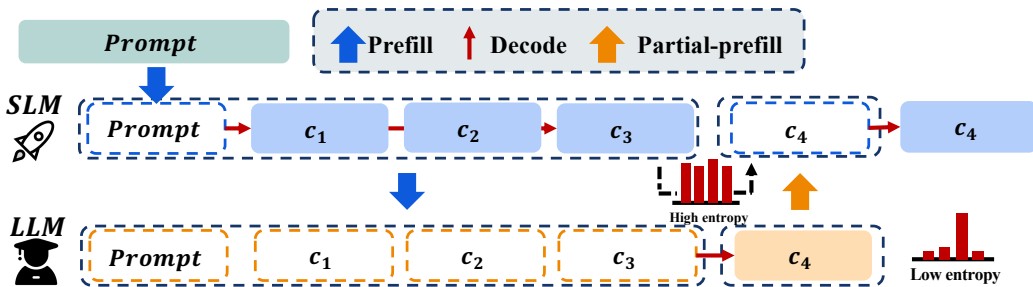

Figure 2: **Overview of R-Stitch.** Given a question with CoT prompting, decoding alternates between an SLM and an LLM under an entropy-based switching policy. Generation starts with the SLM; tokens with low entropy are accepted directly, while high-entropy tokens trigger the LLM to overwrite them and resume decoding. Symmetrically, when the LLM outputs a low-entropy token, it returns to the SLM to reduce computational cost. This bidirectional mechanism adaptively allocates computation, preserving SLM efficiency while leveraging LLM reliability when needed.

## 3 METHODOLOGY

In this section, we build on our empirical finding that high-entropy tokens are more likely to induce errors, and present R-Stitch, a collaborative decoding framework that uses entropy as an uncertainty proxy to coordinate the SLM and LLM. R-Stitch accepts SLM tokens when entropy is low to reduce latency, while delegating high-entropy tokens to the LLM for reliable decoding. This entropy-guided routing exploits the complementary strengths of heterogeneous models and underpins the RL extension R-Stitch$^+$, which learns an adaptive policy beyond fixed thresholds.

### 3.1 PRELIMINARY

**Hybrid decoding setting.** We study a hybrid inference setup with a SLM $f_{\text{SLM}}$ and a LLM $f_{\text{LLM}}$, both autoregressive and sharing the same tokenizer. Given a prompt $x$, the model generates a token sequence $y_{1:T} = [y_1, y_2, \ldots, y_T]$ in an autoregressive manner. At decoding step $t$, the active model $f \in \{f_{\text{SLM}}, f_{\text{LLM}}\}$ outputs a probability distribution $\mathbf{p}_t = f(y_{1:t-1})$, from which the next token $y_t$ is sampled or selected.

### 3.2 R-STITCH: BIDIRECTIONAL ENTROPY-GUIDED DECODING

#### 3.2.1 EMPIRICAL ENTROPY ANALYSIS

LLMs exhibit stronger reasoning capabilities but incur substantially higher inference costs compared to SLMs. Prior work shows that when the two models produce consistent outputs, delegating computation to the SLM can effectively reduce latency. This raises a fundamental question: under what conditions can SLM predictions be relied upon without compromising correctness? To answer this, we conduct sample- and token-level entropy analyses on AMC using DeepSeek-R1-Distill-Qwen-7B (LLM) and L1-1.5B-Short (SLM), which reveal systematic patterns of entropy.

**1. Incorrect answers are associated with higher entropy.** We first examine the link between entropy and reliability. On AMC, the mean entropy over all generated tokens in incorrect reasoning traces is higher than that in correct traces (Figure 3a). In the violin plots, the width of each curve indicates the proportion of tokens falling within a given entropy range, illustrating the distributional difference between correct and incorrect traces. This indicates that *traces with higher average entropy are more likely to lead to incorrect answers, supporting the use of entropy as a practical uncertainty signal.*

**2. Most tokens are generated with extremely low entropy, leaving only a small high-entropy fraction.** We then analyze the overall entropy distribution across reasoning sequences. As shown in Figure 3b, the distribution is heavily skewed toward zero: only 10.65% of SLM-generated tokens exceed an entropy of 0.1, and the majority have entropy exactly 0, indicating very high prediction confidence. This highlights that *high-entropy tokens are relatively rare within a sequence.*

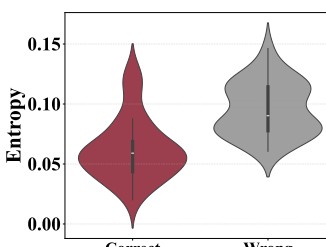 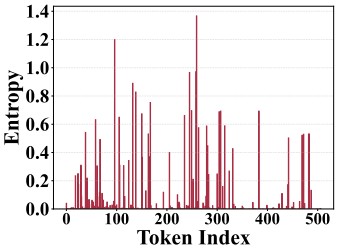 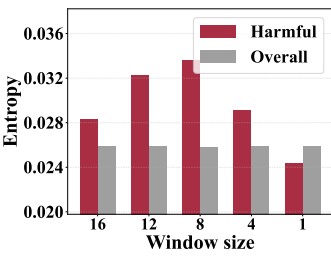

(a) Sample-level entropy in correct vs. incorrect solutions

(b) Token-level entropy distribution across full reasoning traces

(c) Elevated entropy around the first harmful token

Figure 3: **Entropy and error locality.** (a) Incorrect solutions exhibit higher entropy than correct ones. (b) The entropy distribution is heavily skewed toward zero; most tokens have (near-)zero entropy. (c) Neighborhoods around the first harmful token show higher mean entropy than overall entropy distribution, indicating that errors often arise from locally uncertain regions.

**3. High-entropy tokens are more likely to trigger errors.** We investigate the relationship between token entropy and the occurrence of *harmful tokens*—defined as the first SLM tokens whose inclusion flips the LLM's answer from correct to incorrect. This analysis is restricted to AMC problems where the LLM produces the correct solution but the SLM does not. We measure the average entropy in a local window preceding each harmful token and compare it with the global average across all tokens. As shown in Figure 3c, the preceding context of the harmful tokens consistently exhibits higher mean entropy than the overall distribution. This indicates that *high-entropy tokens are more likely to precede and trigger harmful tokens, making them useful signals for routing*.

### 3.2.2 METHOD: BIDIRECTIONAL DECODING WITH ENTROPY ROUTING

Building on these findings, we design R-Stitch to explicitly exploit the observed link between token entropy and error likelihood. R-Stitch is a token-level decoding framework that dynamically alternates between the SLM and LLM based on entropy-derived uncertainty. The key idea is to let the SLM decode as much as possible to reduce latency, while invoking the LLM only when necessary to maintain answer quality. Importantly, the LLM can also transfer control back to the SLM once low-entropy (high-certainty) tokens are reached, enabling a full bidirectional flow. We use a predefined threshold $\tau$ to balance the latency–accuracy trade-off.

**Entropy-guided stitching.** At each decoding step $t$, the active model produces a probability distribution $\mathbf{p}_t$ over the vocabulary of size $V$. We quantify uncertainty by the normalized entropy of this distribution:

$$\mathcal{H}_t = \frac{-\sum_{i=1}^{V} p_{t,i} \log p_{t,i}}{\log V}, \qquad \mathcal{H}_t \in [0,1]. \tag{1}$$

Larger $\mathcal{H}_t$ indicates higher uncertainty, while smaller values indicate higher certainty.

Decoding begins with the SLM. At each step $t$, if the uncertainty $\mathcal{H}_t^{\text{SLM}} \leq \tau$, the SLM accepts its prediction and proceeds. Otherwise ($\mathcal{H}_t^{\text{SLM}} > \tau$), the token is discarded and the step is reprocessed by the LLM, which resumes decoding. Symmetrically, while decoding with the LLM, if $\mathcal{H}_t^{\text{LLM}} \leq \tau$, control is handed back to the SLM; otherwise, the LLM continues.

Formally, the switching policy is:

$$\text{Switch}(t) = \begin{cases} \text{SLM} \to \text{LLM} & \text{if } \mathcal{H}_t^{\text{SLM}} > \tau, \\ \text{LLM} \to \text{SLM} & \text{if } \mathcal{H}_t^{\text{LLM}} \leq \tau. \end{cases} \tag{2}$$

**KV Cache management.** R-Stitch maintains separate key-value caches for the SLM and LLM. When a model first participates in reasoning, it performs a full prefill over the existing input context to initialize its KV cache. Thereafter, each model incrementally updates its own cache during decoding. During model switches, we avoid redundant computation by leveraging *partial prefill*. Specifically, when switching back to a model that has previously decoded, we reuse its existing KV cache and only prefill the new tokens generated by the other model since the last switch. This strategy

eliminates repeated attention over already-processed tokens, thereby enabling efficient cache reuse and significantly reducing switching overhead.

### 3.3 R-STITCH$^+$: RL-BASED ROUTING WITH LATENCY-AWARE REWARD

Beyond entropy-guided stitching, we propose **R-Stitch$^+$**, an RL extension of R-Stitch. At each decoding step, when the token entropy exceeds the threshold $\tau$, the hidden state of the current model is fed into a lightweight router, which decides whether to continue with the SLM or switch to the LLM. Low-entropy tokens are always delegated to the SLM.

**Reward design.** The reward is decomposed into an accuracy term and an efficiency term:

$$R = r_{\text{acc}} + r_{\text{eff}}. \tag{3}$$

The accuracy reward $r_{\text{acc}}$ reflects whether the final prediction is correct, while the efficiency reward $r_{\text{eff}}$ penalizes computational cost. Unlike prior RL approaches for chain-of-thought reasoning that approximate efficiency by the number of generated tokens (Aggarwal & Welleck, 2025), such a proxy is unsuitable here: (i) we involve both a small and a large model, whose per-token costs differ significantly; and (ii) our framework allows partial-prefill operations, whose cost depends jointly on the KV cache size and the length of the newly prefilling span. Ideally, the most faithful measure of efficiency is the actual trajectory latency, so we define

$$r_{\text{eff}} = -\lambda \cdot r_{\text{acc}} \cdot \widehat{L}, \tag{4}$$

where $\widehat{L}$ denotes trajectory latency and $\lambda$ is a trade-off coefficient. Latency is penalized only when the output is correct, ensuring that the router does not pursue speed at the expense of accuracy.

**Latency estimator.** However, directly measuring wall-clock latency during RL rollouts is impractical: trajectories are executed in batches and per-trajectory profiling would incur prohibitive overhead. To address this, we first sample a set of prefilling and decoding operations for each model (SLM and LLM) under different input lengths and KV cache size, and fit their latency profiles via linear regression. Let $N_{\text{inf}}$ denote the number of tokens processed at the current step and $N_{\text{kv}}$ the size of the KV cache before this step. The dominant cost arises from attention, which scales as $\mathcal{O}(N_{\text{inf}} \cdot N_{\text{kv}} + N_{\text{inf}}^2)$, plus linear terms in $N_{\text{inf}}$. Accordingly, we model the latency for a prefill operation as

$$T_{\text{prefill}}(N_{\text{inf}}, N_{\text{kv}}) = a \cdot N_{\text{inf}} \cdot N_{\text{kv}} + b \cdot N_{\text{inf}}^2 + c \cdot N_{\text{inf}} + d, \tag{5}$$

where coefficients $(a, b, c, d)$ are fitted via linear regression on profiling data. A decoding step corresponds to $N_{\text{inf}} = 1$, yielding

$$T_{\text{decode}}(N_{\text{kv}}) = c \cdot N_{\text{kv}} + d. \tag{6}$$

The trajectory-level latency $\widehat{L}$ is then obtained by summing the step-wise estimates.

**DAPO optimization.** We train the router with the DAPO optimizer (Yu et al., 2025). For each prompt $(q, a)$, we sample a group of $G$ routed trajectories $\{o_i\}_{i=1}^{G} \sim \pi_{\theta_{\text{old}}}(\cdot \mid q)$ and compute a scalar reward $R_i$ for each trajectory. Let $o_{i,t}$ be the $t$-th action in $o_i$. We define the importance ratio and the group-normalized advantage as

$$r_{i,t}(\theta) = \frac{\pi_\theta(o_{i,t} \mid q,\, o_{i,<t})}{\pi_{\theta_{\text{old}}}(o_{i,t} \mid q,\, o_{i,<t})}, \hat{A}_{i,t} = \frac{R_i - \text{mean}\big(\{R_j\}_{j=1}^{G}\big)}{\text{std}\big(\{R_j\}_{j=1}^{G}\big)}. \tag{7}$$

Here, $\pi_\theta(\cdot)$ represents the current router policy, while $\pi_{\theta_{\text{old}}}(\cdot)$ denotes the policy used to sample trajectories in the previous iteration. Let $\text{clip}_\varepsilon(x) = \min(\max(x, 1-\varepsilon),\, 1+\varepsilon)$ denote the clipping operator. , the DAPO objective is

$$\mathcal{J}_{\text{DAPO}}(\theta) = \mathbb{E}_{(q,a),\, \{o_i\} \sim \pi_{\theta_{\text{old}}}} \left[ \frac{1}{\sum_{i=1}^{G} |o_i|} \sum_{i=1}^{G} \sum_{t=1}^{|o_i|} \min\Big( r_{i,t}(\theta)\, \hat{A}_{i,t},\ \text{clip}_\varepsilon\big(r_{i,t}(\theta)\big)\, \hat{A}_{i,t} \Big) \right], \tag{8}$$

where $\varepsilon$ is the clipping threshold.

Table 1: Comparison of decoding strategies on five mathematical reasoning datasets. We report accuracy, latency (s/sample), and relative speedup (computed against the corresponding full LLM decoding) under decoding budgets of 8k and 16k tokens. **LLM-7B**, **LLM-14B**, and **LLM-32B** denote DeepSeek-R1-Distill-Qwen-7B, DeepSeek-R1-Distill-Qwen-14B, and QWQ-32B, respectively. **SLM** refers to L1-1.5B-Short (Liu et al., 2025), and **SpecDec** denotes speculative decoding using the corresponding models. Speedup is defined per LLM, per budget, per dataset as Lat(LLM)/Lat(Method).

| Method | $\tau$ | AIME | | | AMC | | | Minerva | | | MATH | | | OlympiadBench | | |
|---|---|---|---|---|---|---|---|---|---|---|---|---|---|---|---|---|
| | | Acc ↑ | Lat. ↓ | Spd. ↑ | Acc ↑ | Lat. ↓ | Spd. ↑ | Acc ↑ | Lat. ↓ | Spd. ↑ | Acc ↑ | Lat. ↓ | Spd. ↑ | Acc ↑ | Lat. ↓ | Spd. ↑ |
| **Decoding budget = 8k tokens** | | | | | | | | | | | | | | | | |
| SLM | – | 10.00 | 5.91 | – | 50.60 | 5.37 | – | 25.37 | 5.03 | – | 73.60 | 4.56 | – | 36.89 | 5.42 | – |
| LLM-7B | – | 33.33 | 86.63 | 1.00× | 66.27 | 63.15 | 1.00× | 31.62 | 41.79 | 1.00× | 86.00 | 38.34 | 1.00× | 51.85 | 117.31 | 1.00× |
| SpecDec | – | 36.67 | 201.23 | 0.43× | 69.88 | 95.42 | 0.66× | 34.19 | 56.59 | 0.74× | 87.00 | 48.71 | 0.79× | 51.85 | 134.23 | 0.87× |
| R-Stitch | 0.001 | 36.67 | 89.86 | 0.96× | 77.11 | 58.72 | 1.08× | 34.19 | 38.73 | 1.08× | 89.40 | 32.94 | 1.16× | 55.26 | 105.29 | 1.11× |
| R-Stitch | 0.02 | 40.00 | 62.03 | 1.40× | 69.88 | 34.89 | 1.81× | 33.09 | 18.98 | 2.20× | 87.00 | 16.61 | 2.31× | 51.85 | 70.43 | 1.67× |
| R-Stitch | 0.03 | 30.00 | 42.06 | 1.84× | 69.88 | 24.55 | 2.57× | 33.09 | 15.07 | 2.77× | 85.60 | 15.15 | 2.53× | 48.59 | 51.43 | 2.28× |
| R-Stitch+ | – | 40.00 | 37.19 | 2.33× | 68.67 | 21.08 | 3.00× | 35.29 | 15.33 | 2.73× | 86.60 | 15.68 | 2.45× | 52.00 | 45.02 | 2.34× |
| LLM-14B | – | 43.33 | 153.20 | 1.00× | 68.67 | 101.76 | 1.00× | 35.29 | 48.02 | 1.00× | 86.00 | 41.66 | 1.00× | 54.07 | 190.89 | 1.00× |
| SpecDec | – | 50.00 | 139.10 | 1.10× | 68.67 | 95.59 | 1.06× | 34.93 | 53.02 | 0.91× | 82.80 | 45.88 | 0.91× | 54.22 | 180.03 | 1.06× |
| R-Stitch | 0.001 | 50.00 | 129.88 | 1.18× | 69.88 | 79.54 | 1.28× | 35.66 | 40.54 | 1.18× | 89.00 | 37.03 | 1.13× | 54.22 | 165.07 | 1.16× |
| R-Stitch | 0.02 | 43.33 | 87.61 | 1.75× | 69.88 | 41.82 | 2.43× | 35.66 | 22.16 | 2.17× | 85.20 | 20.53 | 2.05× | 52.44 | 96.09 | 1.99× |
| R-Stitch | 0.03 | 43.33 | 61.59 | 2.49× | 68.67 | 26.43 | 3.85× | 34.93 | 17.59 | 2.73× | 83.40 | 14.77 | 2.82× | 48.44 | 52.36 | 3.65× |
| R-Stitch+ | – | 50.00 | 107.19 | 1.43× | 73.49 | 49.91 | 2.04× | 35.66 | 26.66 | 1.80× | 88.20 | 29.08 | 1.43× | 56.30 | 108.67 | 1.76× |
| LLM-32B | – | 43.33 | 354.85 | 1.00× | 60.24 | 292.78 | 1.00× | 41.18 | 231.10 | 1.00× | 87.53 | 178.38 | 1.00× | 50.67 | 379.52 | 1.00× |
| SpecDec | – | 40.00 | 270.25 | 1.31× | 59.04 | 209.72 | 1.40× | 41.91 | 182.28 | 1.27× | 88.60 | 136.73 | 1.30× | 50.04 | 351.27 | 1.08× |
| R-Stitch | 0.001 | 50.00 | 261.86 | 1.40× | 68.75 | 209.88 | 1.39× | 42.65 | 141.24 | 1.64× | 91.20 | 95.49 | 1.87× | 50.67 | 341.67 | 1.11× |
| R-Stitch | 0.02 | 50.00 | 184.97 | 1.92× | 68.67 | 118.26 | 2.48× | 36.76 | 59.58 | 3.87× | 89.80 | 43.49 | 4.10× | 53.78 | 226.71 | 1.67× |
| R-Stitch | 0.03 | 40.00 | 178.19 | 1.99× | 69.88 | 86.88 | 3.37× | 34.56 | 43.41 | 5.32× | 87.00 | 32.21 | 5.54× | 52.15 | 157.47 | 2.41× |
| **Decoding budget = 16k tokens** | | | | | | | | | | | | | | | | |
| SLM | – | 10.00 | 5.91 | – | 50.60 | 5.37 | – | 25.37 | 5.03 | – | 73.60 | 4.56 | – | 36.89 | 5.42 | – |
| LLM-7B | – | 40.00 | 195.77 | 1.00× | 71.08 | 116.41 | 1.00× | 34.93 | 54.43 | 1.00× | 90.80 | 50.07 | 1.00× | 58.67 | 208.77 | 1.00× |
| SpecDec | – | 50.00 | 258.54 | 0.76× | 80.72 | 132.45 | 0.88× | 35.66 | 115.76 | 0.47× | 91.20 | 44.47 | 1.13× | 60.15 | 267.03 | 0.78× |
| R-Stitch | 0.001 | 46.67 | 192.22 | 1.03× | 80.72 | 97.26 | 1.20× | 35.66 | 45.53 | 1.20× | 91.00 | 38.34 | 1.31× | 60.15 | 201.13 | 1.04× |
| R-Stitch | 0.02 | 50.00 | 110.16 | 1.78× | 67.47 | 54.17 | 2.15× | 33.09 | 18.79 | 2.90× | 88.60 | 22.56 | 2.22× | 58.67 | 109.98 | 1.90× |
| R-Stitch | 0.03 | 36.67 | 41.51 | 4.72× | 66.27 | 38.07 | 3.06× | 35.29 | 13.50 | 4.03× | 83.20 | 15.62 | 3.21× | 54.22 | 65.65 | 3.18× |
| R-Stitch+ | – | 50.00 | 86.03 | 2.28× | 77.11 | 56.03 | 2.08× | 35.29 | 13.68 | 3.98× | 88.60 | 16.02 | 3.13× | 59.11 | 95.31 | 2.19× |
| LLM-14B | – | 50.00 | 246.95 | 1.00× | 86.75 | 146.51 | 1.00× | 39.34 | 68.99 | 1.00× | 88.60 | 66.54 | 1.00× | 60.00 | 316.06 | 1.00× |
| SpecDec | – | 50.00 | 275.64 | 0.90× | 83.13 | 138.53 | 1.06× | 39.34 | 55.40 | 1.25× | 87.40 | 67.18 | 0.99× | 60.00 | 360.14 | 1.05× |
| R-Stitch | 0.001 | 56.67 | 217.06 | 1.14× | 79.52 | 90.72 | 1.61× | 37.87 | 40.46 | 1.70× | 89.40 | 39.82 | 1.67× | 61.06 | 290.20 | 1.09× |
| R-Stitch | 0.02 | 43.33 | 99.12 | 2.49× | 66.27 | 51.29 | 2.86× | 31.99 | 22.74 | 3.04× | 87.80 | 22.62 | 2.94× | 54.22 | 108.61 | 2.91× |
| R-Stitch | 0.03 | 40.00 | 54.89 | 4.50× | 63.86 | 28.38 | 5.16× | 33.82 | 16.14 | 4.27× | 84.80 | 17.37 | 3.83× | 48.59 | 68.63 | 4.61× |
| R-Stitch+ | – | 53.33 | 132.98 | 1.86× | 79.52 | 68.50 | 2.14× | 37.13 | 35.11 | 1.96× | 89.60 | 33.70 | 1.97× | 59.11 | 121.39 | 2.60× |
| LLM-32B | – | 56.67 | 591.67 | 1.00× | 87.95 | 419.94 | 1.00× | 46.69 | 348.48 | 1.00× | 94.00 | 249.92 | 1.00× | 67.70 | 722.77 | 1.00× |
| SpecDec | – | 70.00 | 526.03 | 1.12× | 87.95 | 326.73 | 1.29× | 44.23 | 273.07 | 1.28× | 93.20 | 187.79 | 1.33× | 67.70 | 798.23 | 0.90× |
| R-Stitch | 0.001 | 70.00 | 488.87 | 1.21× | 90.36 | 288.92 | 1.45× | 41.18 | 126.62 | 2.75× | 94.00 | 172.08 | 1.45× | 66.07 | 654.24 | 1.10× |
| R-Stitch | 0.02 | 50.00 | 333.17 | 1.78× | 81.93 | 168.19 | 2.50× | 40.07 | 73.32 | 4.75× | 90.60 | 80.21 | 3.12× | 61.19 | 358.87 | 2.01× |
| R-Stitch | 0.03 | 53.33 | 276.03 | 2.14× | 73.49 | 143.53 | 2.93× | 36.40 | 43.58 | 8.00× | 87.80 | 38.32 | 6.52× | 55.70 | 209.96 | 3.44× |

## 4 EXPERIMENTS

**Implementation details.** We evaluate our proposed method on five mathematical reasoning benchmarks: OlympiadBench (He et al., 2024a), AIME (Li et al., 2024a), Minerva (Lewkowycz et al., 2022), AMC (Li et al., 2024a), and MATH (Hendrycks et al., 2021). We adopt DeepSeek-Math-R1-Distill-Qwen models at 7B and 14B scales, and QwQ-32B as the 32B-scale LLM, with L1-Short serving as the SLM. Our method is implemented within the vLLM (Kwon et al., 2023) inference framework, where the LLM and SLM are instantiated as separate engines, each maintaining its own key-value cache to enable independent decoding and efficient model switching. Speculative decoding baselines are evaluated using the official vLLM speculative decoding code. For R-Stitch+, we train the router using DAPO (Yu et al., 2025)'s publicly released RL training dataset, with $\lambda = 5 \times 10^{-6}$, $\tau = 0.001$, batch size 32, and rollout group size 8. The latency estimator is calibrated in milliseconds. Due to GPU memory constraints, RL experiments are conducted only on the 7B and 14B models. All latency numbers reported in the results correspond to the average wall-clock inference time per sample on a single NVIDIA A100 GPU with batch size 1.

**Main results on math reasoning benchmarks.** We evaluate R-Stitch under entropy thresholds $\tau \in \{0.001, 0.02, 0.03\}$, comparing against SLM-only decoding, full LLM decoding, and speculative decoding (SpecDec). Table 1 reports accuracy, latency, and relative speedup across five mathematical reasoning benchmarks under decoding budgets of 8k and 16k tokens. Overall, R-Stitch consistently reduces latency with minimal accuracy degradation relative to full LLM decoding. At $\tau = 0.02$, LLM-7B achieves $1.4\times$–$2.3\times$ speedups at 8k and $1.7\times$–$2.9\times$ at 16k, while maintaining accuracy close to the LLM baseline. For larger models, efficiency gains become more pronounced: LLM-14B reaches $1.7\times$–$2.4\times$ at 8k and up to $3.0\times$ at 16k, while LLM-32B attains speedups of $1.9\times$–$4.1\times$

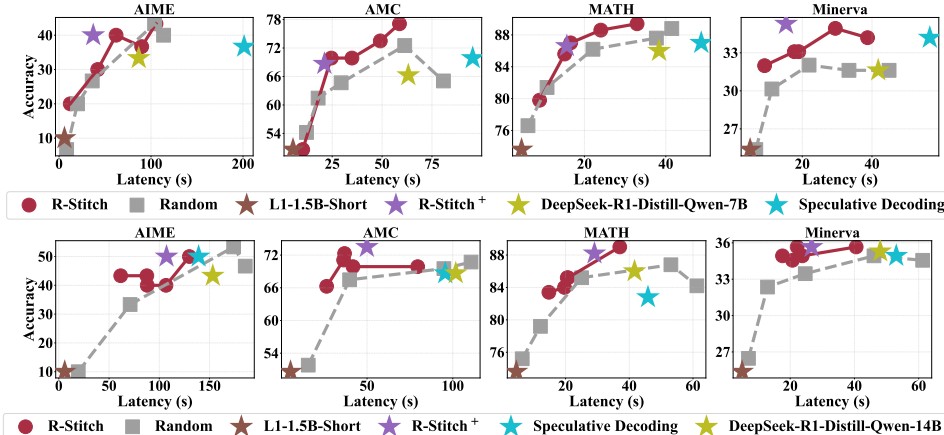

Figure 4: Accuracy–latency trade-off curves on mathematical reasoning datasets. The first row uses the 7B LLM, and the second row uses the 14B LLM. The red lines correspond to our method (R-Stitch) under varying entropy thresholds $\tau$ and a random routing baseline.

at 8k and $1.8\times-4.8\times$ at 16k. In particular, the observed accuracy gains arise because R-Stitch shortens overly verbose LLM reasoning traces, preventing truncation under strict length budgets and thereby avoiding accuracy degradation. SpecDec shows mixed performance: while occasionally competitive, it often runs slower than standard LLM decoding (e.g., 7B at 8k) or sacrifices accuracy. This slowdown is largely due to frequent rejections under low model agreement, which cause the tokens of SLM to be discarded and re-decoded by the LLM, resulting in more total decoding steps than even vanilla LLM decoding. SLM-only decoding remains fastest but yields substantially lower accuracy, highlighting the necessity of hybrid approaches. These results demonstrate that entropy-based routing provides a robust trade-off between efficiency and accuracy across different model scales and token budgets, with larger models benefiting the most in relative speedups.

**Latency–accuracy trade-off.** We further assess entropy-guided routing by varying the threshold $\tau \in \{0.001, 0.005, 0.02, 0.03, 0.05\}$ and reporting the resulting accuracy–latency trade-off. As baselines, we consider random routing, where tokens are assigned to the SLM with fixed probabilities $p \in \{0.1, 0.3, 0.5, 0.7, 0.9\}$, along with fixed decoding strategies including SLM-only, LLM-only, and speculative decoding. Figure 4 presents results on four mathematical reasoning benchmarks with both 7B and 14B LLMs. Across all settings, R-Stitchconsistently outperforms random routing, confirming entropy as a reliable confidence signal. On relatively simple tasks such as AMC, it achieves accuracy comparable to full LLM decoding while operating at substantially lower latency. Overall, entropy-based routing offers a simple, training-free mechanism that effectively balances efficiency and accuracy across diverse datasets and model scales. In addition, by tuning the entropy threshold, R-Stitchnaturally adapts to different computational budgets without retraining, enabling practitioners to approach optimal accuracy under varying latency constraints.

**Adaptive collaboration is important.** In Table 1, both methods are compared under the setting where the LLM is paired with a relatively low-consistency SLM. For Figure 5, however, we intentionally assign speculative decoding a more favorable pairing by using a high-consistency distilled model, since speculative decoding relies on model agreement to function effectively. This ensures that speculative decoding is evaluated under its better condition, rather than being penalized by mismatch. Even so, its acceleration remains limited, as it inherits the verbosity of the LLM. In contrast, R-Stitch is able to exploit concise but less consistent models such as L1-1.5B-Short, yielding about 50% fewer tokens and substantially lower latency. *These results highlight that adaptive token-level collaboration between heterogeneous models is essential to fully exploit their complementary strengths.* More detailed statistics are provided in the appendix.

**Per-sample comparison with Speculative Decoding.** Figure 6 provides per-sample visualizations of latency speedup and token reduction when applying R-Stitch[+] and Speculative Decoding on the LLM-7B in AMC. We observe that R-Stitch[+] consistently accelerates the majority of samples, while Speculative Decoding only yields speedup on a small portion of samples and causes slowdowns on most due to consistency issues. Notably, the theoretical upper bound of speculative decoding's

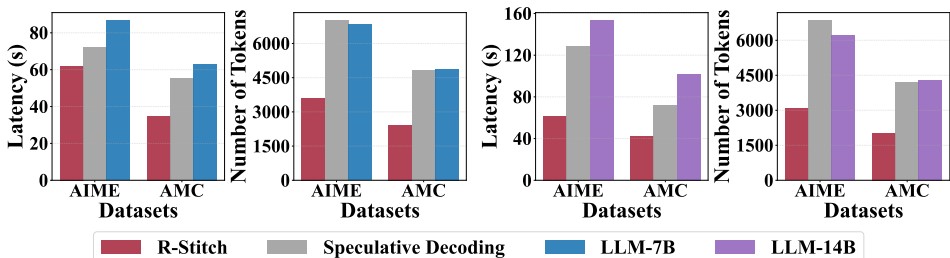

Figure 5: **Comparison with full LLM decoding and speculative decoding using high-consistency pairs.** Both R-Stitch and speculative decoding maintain the accuracy of full LLM decoding while reducing latency. Speculative decoding pairs the LLM with a high-consistency distilled model (R1-Distill-Qwen-1.5B), whereas R-Stitch employs the more concise but less consistent L1-1.5B-Short.

speedup under high consistency is the latency ratio between the LLM and SLM. In the visualization, the fastest sample achieves just above a $2\times$ speedup, which matches the decoding latency gap between the 7B and 1.5B models. *In contrast, our method additionally leverages the concise expressiveness of the SLM while preserving accuracy, thereby achieving substantially higher acceleration of up to* ***14×***.

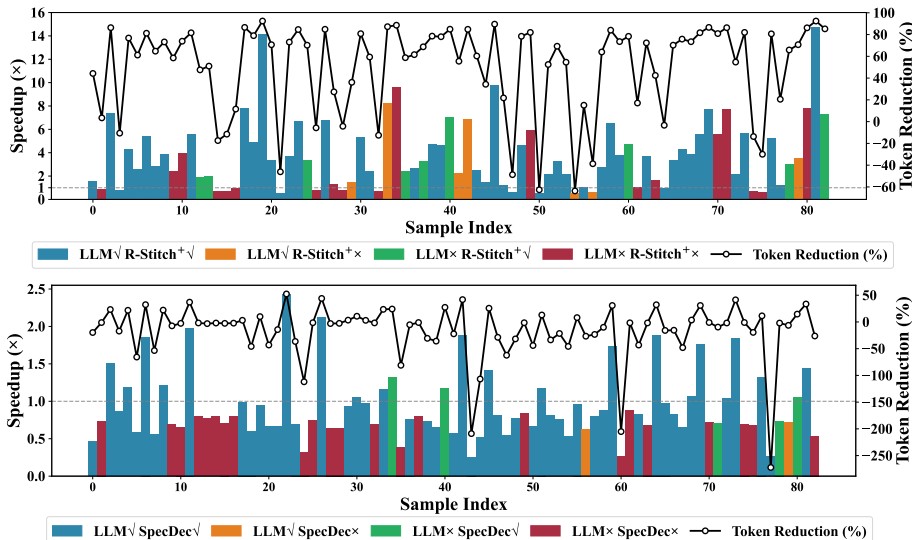

Figure 6: **Per-sample visualization of R-Stitch$^+$ and Speculative Decoding (LLM-7B).** Each bar shows the latency speedup of one sample relative to the baseline LLM. Bar colors encode correctness outcomes of the baseline and R-Stitch$^+$. The dashed horizontal line at 1 indicates no speedup. The black solid curve with hollow-circle markers represents token reduction percentages per sample.

## 5 CONCLUSION

This paper has introduced R-Stitch, a dynamic routing strategy for accelerating large language model inference by selectively delegating token-level computation to a small language model (SLM). Our method leverages an entropy-based switching mechanism that routes easy tokens to the lightweight model while preserving overall output quality. This design enables a favorable trade-off between inference latency and accuracy without requiring additional retraining or architectural changes. Furthermore, we have extended this approach to R-Stitch$^+$, which incorporates a reinforcement learning (RL)–based router trained with a latency-aware reward. By adaptively deciding when to switch between the SLM and the LLM, R-Stitch$^+$ refines the efficiency–accuracy balance beyond fixed entropy thresholds, achieving stronger acceleration while maintaining high accuracy. Comprehensive experiments on multiple mathematical reasoning and code generation datasets have demonstrated that our framework achieves consistent improvements over static baselines and speculative decoding, offering a practical and efficient solution for real-world deployment of LLMs.

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

# A Appendix

## A.1 Use of Large Language Models

We used GPT-based language models solely for writing assistance, including grammar correction and stylistic refinement of sentences. No LLMs were involved in the design of methods and experiments.

## A.2 Limitations and future work

Our current implementation supports only batch size 1 due to dynamic token-level model switching, which restricts hardware utilization in practical deployment. Addressing this limitation may require designing new scheduling strategies or restructuring the routing mechanism to better accommodate batched inference. To further alleviate the KV cache and parameter burden from maintaining two models, we plan to adopt parameter-sharing strategies such as mixture-of-depth/width to enhance memory efficiency. These extensions could improve the robustness and scalability of dynamic model routing, and enable more fine-grained control over the latency–accuracy trade-off.

## A.3 Inference algorithm of the proposed method

---

**Algorithm 1** Inference procedure of the proposed method

---

**Require:** Prompt $x$; models SLM, LLM; threshold $\tau$; max length $T_{\max}$
1: Optional: router $\pi_\theta$ (R-Stitch$^+$)
2: $\text{KV}^{\text{SLM}} \leftarrow \emptyset$, $\text{KV}^{\text{LLM}} \leftarrow \emptyset$; $L^{\text{SLM}} = L^{\text{LLM}} = 0$  $\triangleright$ KV$^{\text{SLM}}$, KV$^{\text{LLM}}$: KV caches; $L^{\text{SLM}}$, $L^{\text{LLM}}$: KV cache
   sizes of SLM and LLM
3: $y \leftarrow []$, ACTIVE $\leftarrow$ SLM  $\triangleright$ $y$: generated output; ACTIVE: current inference model
4: **for** $t = 1$ **to** $T_{\max}$ **do**
5:   **if** $L^{\text{ACTIVE}} < |x \oplus y|$ **then**
6:     **Prefill**: run ACTIVE on $(x \oplus y)[L^{\text{ACTIVE}}:]$ to update cache and generate $(\mathbf{p}_t, y_t)$; $L^{\text{ACTIVE}} \leftarrow |x \oplus y| + 1$
        $\triangleright$ KV cache not up-to-date; $\mathbf{p}_t$: prob. distribution; $y_t$: generated token
7:   **else**
8:     **Decode**: run ACTIVE with cache to generate $(\mathbf{p}_t, y_t)$; $L^{\text{ACTIVE}} \leftarrow L^{\text{ACTIVE}} + 1$
9:   **end if**
10:  Compute entropy $\mathcal{H}_t$ from $\mathbf{p}_t$ (Eq. 1)  $\triangleright$ $\mathcal{H}_t$: token entropy
11:  **if** $\mathcal{H}_t > \tau$ **then**
12:    **if** R-Stitch **then**
13:      SWITCH $\leftarrow$ True
14:    **else if** R-Stitch$^+$ **then**
15:      SWITCH $\leftarrow (\arg\max_{a \in \{\text{SLM},\text{LLM}\}} \pi_\theta(a \mid s_t) = \text{LLM})$  $\triangleright$ router decision (state $s_t$)
16:    **end if**
17:    **if** ACTIVE = SLM **and** SWITCH **then**
18:      **Discard** $y_t$ and rollback KV$^{\text{SLM}}$; ACTIVE $\leftarrow$ LLM; **continue**  $\triangleright$ switch SLM→LLM, token not
         kept
19:    **end if**
20:  **else if** ACTIVE = LLM **then**
21:    ACTIVE $\leftarrow$ SLM  $\triangleright$ switch LLM→SLM
22:  **end if**
23:  Append $y_t$ to $y$; update cache  $\triangleright$ append only if token not discarded
24:  **if** $y_t = $ [EOS] **then**
25:    **break**
26:  **end if**
27: **end for**
28: **return** $y$

---

## A.4 Latency Regression Results

Building on the latency estimator introduced in Eq. 5 and Eq. 6, we report the fitted coefficients $(a, b, c, d)$ obtained from profiling three representative models: SLM-1.5B, LLM-7B, and LLM-14B. Recall that $N_{\text{inf}}$ denotes the number of tokens processed at the current step and $N_{\text{kv}}$ the size of the KV cache before this step. Latency is modeled as

$$T(N_{\text{inf}}, N_{\text{kv}}) = a \cdot N_{\text{inf}} \cdot N_{\text{kv}} + b \cdot N_{\text{inf}}^2 + c \cdot N_{\text{inf}} + d.$$

The fitted functions are as follows (latency $T$ in milliseconds):

- SLM-1.5B:

$$T(N_{\text{inf}}, N_{\text{kv}}) = 0.000021 \cdot (N_{\text{inf}} \cdot N_{\text{kv}}) + 0.000231 \cdot N_{\text{inf}}^2 - 0.121046 \cdot N_{\text{inf}} + 27.090929$$

- LLM-7B:

$$T(N_{\text{inf}}, N_{\text{kv}}) = 0.000027 \cdot (N_{\text{inf}} \cdot N_{\text{kv}}) + 0.000031 \cdot N_{\text{inf}}^2 - 0.045256 \cdot N_{\text{inf}} + 27.040801$$

- LLM-14B:

$$T(N_{\text{inf}}, N_{\text{kv}}) = 0.000045 \cdot (N_{\text{inf}} \cdot N_{\text{kv}}) + 0.000123 \cdot N_{\text{inf}}^2 - 0.082998 \cdot N_{\text{inf}} + 45.118931$$

These regression results provide practical estimators for per-step latency under varying input lengths and KV cache sizes, enabling efficient evaluation without costly profiling.

### A.5 RELATED WORK OF OPTIMIZING SPECULATIVE DECODING

Recent approaches to speculative decoding acceleration can be understood as attempts to train auxiliary mechanisms that increase the consistency between the draft model and the target model. This line of research is exemplified by the EAGLE family and Griffin, while Hydra represents a lighter-weight but still consistency-oriented variant, and CITER departs somewhat from this trend but nevertheless introduces additional training cost. We briefly review these methods below.

**EAGLE.** The EAGLE series (Li et al., 2024b;c; 2025b) develops progressively more sophisticated training strategies for draft models. The original EAGLE (Li et al., 2024b) proposed to autoregressively predict the top-layer hidden features of the target model, rather than tokens directly, thereby anchoring the draft model's outputs more closely to the target distribution. This was combined with a tree-based drafting mechanism and corresponding tree attention to improve parallelism. EAGLE-2 (Li et al., 2024c) extended this by introducing a dynamic draft tree that adapts its expansion based on the draft model's confidence, reducing wasted computation on unlikely branches. Finally, EAGLE-3 (Li et al., 2025b) removed the restriction of feature-level prediction and returned to token-level generation, but enhanced it with multi-level feature fusion and a novel training-time test technique, which simulates self-feedback during training. These changes allow EAGLE-3 to scale better with more training data and to reach higher acceptance rates, but all variants require substantial supervised training of the draft model.

**Griffin.** Griffin (Hu et al., 2025) addresses the bottleneck of speculative decoding by improving the quality of the small draft model rather than modifying the large model or the verification process. It introduces guidance distillation to better align the draft model's predictive distribution with that of the target model, and augments the architecture with lightweight enhancements that increase expressiveness without incurring significant cost. As a result, Griffin achieves substantially higher acceptance rates than naive draft models, thereby unlocking more consistent acceleration for speculative decoding.

**Hydra.** Hydra (Ankner et al., 2024) departs from training a separate small model and instead equips the large model with a set of Hydra heads, lightweight proposal modules attached to intermediate hidden states. Unlike the earlier Medusa framework (Cai et al., 2024), where draft heads were independent and predicted tokens in parallel, Hydra enforces sequential dependency across heads: each head conditions on the tokens predicted by previous heads. This design increases the internal consistency of the drafted sequence, leading to higher acceptance rates without the need for an external small model. Hydra thus offers a lighter-weight solution, but it still shares the central assumption that greater draft–target consistency is the key to efficiency.

**Distinctive perspective of our method.** In contrast to all these approaches, our work takes a fundamentally different perspective. We do not attempt to make the small and large models more consistent, nor do we introduce training overhead for rewriting. Instead, we explicitly exploit the insight that the small model is *simpler and naturally inconsistent* with the large model. Rather than mitigating this inconsistency, we turn it into an advantage, showing that it can be directly leveraged to improve speculative decoding efficiency. Training-free R-Stitch requires *no additional training* and no auxiliary modules, yet achieves significant acceleration. This reveals a new perspective: sometimes inconsistency between models is not a limitation to be avoided, but a property that can be actively harnessed for efficiency gains.

Table 2: Performance comparison with the state-of-the-art speculative decoding baseline (Eagle-3, the fastest method reported in the official vLLM repository) on Qwen3-8/14B (Yang et al., 2025a) across four mathematical reasoning benchmarks. We report accuracy (Acc), latency (s/sample), average output length (Tok.), and relative speedup (Spd.), under both 8k-token and 16k-token decoding budgets.

| Method | $\tau$ | AIME | | | | AMC | | | | Minerva | | | | MATH | | | |
|---|---|---|---|---|---|---|---|---|---|---|---|---|---|---|---|---|---|
| | | Acc ↑ | Lat. ↓ | Tok. ↓ | Spd. ↑ | Acc ↑ | Lat. ↓ | Tok. ↓ | Spd. ↑ | Acc ↑ | Lat. ↓ | Tok. ↓ | Spd. ↑ | Acc ↑ | Lat. ↓ | Tok. ↓ | Spd. ↑ |
| Decoding budget = 8k tokens | | | | | | | | | | | | | | | | | |
| Qwen3-8B | – | 30.00 | 97.50 | 7589.33 | 1.00× | 59.04 | 81.68 | 6384.39 | 1.00× | 34.93 | 68.49 | 5367.97 | 1.00× | 84.00 | 56.02 | 4423.09 | 1.00× |
| Eagle-3 | – | 36.67 | 64.87 | 7740.33 | 1.50× | 59.04 | 52.35 | 6575.19 | 1.56× | 36.03 | 42.30 | 5380.57 | 1.62× | 86.00 | 33.87 | 4437.59 | 1.65× |
| R-Stitch | 0.001 | 50.00 | 64.89 | 6978.97 | 1.50× | 71.08 | 44.72 | 4942.05 | 1.83× | 42.65 | 35.68 | 3913.13 | 1.92× | 89.60 | 24.77 | 2845.66 | 2.26× |
| R-Stitch | 0.01 | 56.67 | 48.30 | 5527.50 | 2.02× | 62.65 | 27.53 | 3316.34 | 2.97× | 38.97 | 27.53 | 2030.41 | 2.49× | 87.00 | 12.52 | 1614.31 | 4.47× |
| R-Stitch | 0.02 | 26.67 | 39.35 | 4701.90 | 2.48× | 66.27 | 19.27 | 2453.30 | 2.97× | 34.56 | 10.86 | 1416.57 | 6.31× | 83.60 | 9.27 | 1260.76 | 6.04× |
| Qwen3-14B | – | 40.00 | 158.80 | 7549.47 | 1.00× | 62.65 | 126.86 | 6055.57 | 1.00× | 41.54 | 100.52 | 4808.01 | 1.00× | 89.00 | 82.93 | 3980.87 | 1.00× |
| Eagle-3 | – | 43.33 | 110.48 | 7700.53 | 1.44× | 60.24 | 83.70 | 6125.43 | 1.52× | 39.71 | 67.10 | 4891.85 | 1.50× | 88.20 | 53.89 | 4056.89 | 1.54× |
| R-Stitch | 0.001 | 46.67 | 94.97 | 7043.47 | 1.67× | 67.47 | 63.13 | 4802.56 | 2.01× | 45.59 | 46.30 | 3545.74 | 2.17× | 90.00 | 32.62 | 2688.23 | 2.54× |
| R-Stitch | 0.01 | 43.33 | 75.51 | 6213.47 | 2.10× | 65.06 | 35.75 | 3213.11 | 3.55× | 40.44 | 20.53 | 1815.11 | 4.90× | 88.00 | 16.24 | 1569.15 | 5.11× |
| R-Stitch | 0.02 | 33.33 | 55.20 | 4836.30 | 2.88× | 61.45 | 23.99 | 2330.29 | 5.29× | 37.13 | 13.01 | 1266.13 | 7.73× | 84.60 | 11.64 | 1223.35 | 7.12× |
| Decoding budget = 16k tokens | | | | | | | | | | | | | | | | | |
| Qwen3-8B | – | 63.33 | 156.21 | 11899.80 | 1.00× | 81.93 | 115.33 | 9022.71 | 1.00× | 47.43 | 90.28 | 6822.03 | 1.00× | 94.40 | 65.05 | 5067.67 | 1.00× |
| Eagle-3 | – | 66.67 | 110.23 | 12080.73 | 1.42× | 81.93 | 76.55 | 9008.47 | 1.51× | 47.06 | 58.49 | 6981.51 | 1.54× | 95.60 | 40.60 | 5150.57 | 1.60× |
| R-Stitch | 0.0005 | 70.00 | 109.82 | 10942.50 | 1.42× | 81.93 | 70.46 | 7107.68 | 1.64× | 46.32 | 48.65 | 3640.69 | 1.86× | 94.60 | 33.64 | 3640.69 | 1.93× |
| R-Stitch | 0.001 | 60.00 | 102.83 | 10135.03 | 1.52× | 74.70 | 65.14 | 6606.36 | 1.77× | 41.54 | 44.73 | 4510.72 | 2.02× | 94.80 | 29.20 | 3238.44 | 2.23× |
| R-Stitch | 0.01 | 53.33 | 69.63 | 7506.93 | 2.24× | 73.49 | 38.13 | 4319.67 | 3.02× | 37.50 | 18.04 | 2142.44 | 5.00× | 89.60 | 14.97 | 1839.84 | 4.35× |
| Qwen3-14B | – | 70.00 | 248.17 | 11237.64 | 1.00× | 81.93 | 167.32 | 7896.86 | 1.00× | 46.69 | 125.74 | 5959.92 | 1.00× | 95.60 | 94.76 | 4528.12 | 1.00× |
| Eagle-3 | – | 63.33 | 189.59 | 11203.67 | 1.31× | 85.54 | 121.30 | 8292.41 | 1.38× | 47.06 | 84.50 | 5930.95 | 1.49× | 95.40 | 63.31 | 4616.20 | 1.50× |
| R-Stitch | 0.0005 | 70.00 | 139.11 | 9637.84 | 1.78× | 85.54 | 89.25 | 6346.75 | 1.87× | 45.96 | 59.48 | 4373.63 | 2.11× | 94.60 | 45.60 | 3427.84 | 2.08× |
| R-Stitch | 0.001 | 70.00 | 136.51 | 9556.70 | 1.82× | 83.13 | 91.11 | 6713.41 | 1.84× | 44.85 | 53.10 | 3985.24 | 2.37× | 94.20 | 37.34 | 3003.98 | 2.54× |
| R-Stitch | 0.01 | 56.67 | 113.50 | 8668.10 | 2.19× | 71.08 | 40.84 | 3512.52 | 4.10× | 38.24 | 24.58 | 2116.48 | 5.12× | 88.60 | 20.65 | 1868.24 | 4.59× |

## A.6 COMPARISON WITH STATE-OF-THE-ART SPECULATIVE DECODING

Table 2 compares our method with Eagle-3, the state-of-the-art speculative decoding approach reported in the official vLLM repository, on Qwen3-8B and Qwen3-14B across four mathematical reasoning benchmarks under both 8k and 16k decoding budgets. Both Eagle-3 and R-Stitch exploit the advantage that small models decode tokens faster than the LLM, thereby reducing per-token complexity. However, R-Stitch additionally leverages the conciseness of SLM outputs: by selectively retaining short, low-entropy continuations from the SLM, it reduces the overall output length, yielding substantially higher end-to-end acceleration. Notably, Eagle-3 employs a heavily trained draft model much smaller than our 1.5B SLM, making its per-token decoding inherently faster. Despite this disadvantage, R-Stitch achieves consistently larger speedups across all benchmarks and budgets, highlighting the benefit of flexible entropy-guided collaboration over purely consistency-driven speculative decoding.

## A.7 COMBINING WITH EARLY EXIT STRATEGIES

Table 3: Effect of applying R-Stitch ($\tau = 0.01$) on top of DEER with the LLM-7B under a 16k-token decoding budget. Accuracy (%), average token count, and inference latency (s) are reported. Results of the vanilla LLM-7B are also included for reference.

| Dataset | LLM-7B | | | + DEER | | | + DEER + R-Stitch | | |
|---|---|---|---|---|---|---|---|---|---|
| | Acc. (%) | Token | Lat. (s) | Acc. (%) | Token | Lat. (s) | Acc. (%) | Token | Lat. (s) |
| MATH | 90.80 | 3381.51 | 50.07 | 89.20 | 2284.52 | 35.47 | 88.40 | 1564.90 | 20.59 |
| AIME | 40.00 | 11739.10 | 195.77 | 36.67 | 9229.03 | 210.37 | 36.67 | 4425.20 | 92.52 |
| GSM8K | 88.93 | 1398.54 | 20.91 | 90.07 | 698.47 | 9.69 | 90.07 | 462.41 | 6.76 |
| GPQA-D | 23.74 | 10134.81 | 171.76 | 27.78 | 7084.15 | 117.50 | 30.81 | 1047.85 | 18.15 |

We further examine whether R-Stitch can complement training-free early-exit methods to improve decoding efficiency. Specifically, we combine R-Stitch with DEER (Yang et al., 2025b), which halts generation once the model's confidence surpasses a predefined threshold. As shown in Table 3, evaluated on MATH (Hendrycks et al., 2021), AIME (Li et al., 2024a), GSM8K (Cobbe et al., 2021), and GPQA-D (Rein et al., 2024) under a 16k-token budget, the hybrid system substantially reduces both token count and end-to-end latency compared to DEER alone, while maintaining comparable accuracy. On AIME, token usage is reduced by more than 50% and latency drops from 210.37s to 92.52s, without affecting accuracy. On GPQA-D, the benefits are even more pronounced: latency decreases from 117.50s to 18.15s, while accuracy improves from 27.78% to 30.81%. Compared to

vanilla LLM-7B decoding (171.76s), this corresponds to a **9.5×** acceleration. These results confirm that early exiting and entropy-guided routing are complementary: DEER shortens the reasoning trajectory, whereas R-Stitch lowers per-token cost by selectively invoking the LLM only when needed. Together, they provide strictly greater efficiency by simultaneously addressing sequence length and per-step computation. For fairness, both +DEER and +DEER+R-Stitch results are reproduced using the official vLLM-based implementation of DEER, where the full context is re-prefilled after each early-exit decision. This design substantially underestimates the achievable efficiency gains, limiting the apparent acceleration of both DEER alone and our combined method. Our integration retains DEER's stopping criterion but substitutes its single-model decoding with R-Stitch, ensuring that the reported improvements are attributable to our method.

Table 4: Comparison of decoding strategies on three code generation benchmarks. We report accuracy, latency (s/sample), and relative speedup (computed against the corresponding full LLM decoding). LLM-7B and LLM-14B denote DeepSeek-R1-Distill-Qwen-7B and DeepSeek-R1-Distill-Qwen-14B, respectively. SLM refers to L1-1.5B-Short (Liu et al., 2025), and SpecDec denotes speculative decoding using the corresponding LLM.

| Method | $\tau$ | LiveCodeBench | | | MBPP | | | HumanEval | | | **Average** | | |
|---|---|---|---|---|---|---|---|---|---|---|---|---|---|
| | | Acc ↑ | Lat. ↓ | Spd. ↑ | Acc ↑ | Lat. ↓ | Spd. ↑ | Acc ↑ | Lat. ↓ | Spd. ↑ | Acc ↑ | Lat. ↓ | Spd. ↑ |
| SLM | – | 8.81 | 9.74 | – | 27.00 | 1.45 | – | 42.70 | 2.26 | – | 26.17 | 4.48 | – |
| LLM-7B | – | 40.90 | 92.46 | 1.00× | 64.00 | 23.97 | 1.00× | 78.60 | 38.97 | 1.00× | 61.17 | 51.80 | 1.00× |
| SpecDec | – | 40.31 | 112.21 | 0.82× | 62.40 | 18.72 | 1.28× | 80.49 | 47.83 | 0.81× | 61.07 | 59.59 | 0.87× |
| R-Stitch | 0.001 | 40.31 | 87.67 | 1.05× | 59.30 | 19.62 | 1.22× | 74.40 | 22.32 | 1.75× | 58.00 | 43.20 | 1.20× |
| R-Stitch | 0.005 | 41.88 | 82.12 | 1.07× | 48.90 | 13.65 | 1.76× | 67.70 | 14.18 | 2.75× | 52.83 | 36.65 | 1.41× |
| R-Stitch | 0.01 | 36.59 | 78.54 | 1.18× | 49.70 | 10.98 | 2.18× | 62.80 | 7.46 | 5.22× | 49.70 | 32.33 | 1.60× |
| LLM-14B | – | 49.51 | 183.26 | 1.00× | 74.90 | 7.52 | 1.00× | 86.00 | 25.10 | 1.00× | 70.14 | 71.96 | 1.00× |
| SpecDec | – | 42.27 | 234.62 | 0.78× | 72.20 | 4.25 | 1.77× | 78.60 | 16.92 | 1.48× | 64.36 | 85.26 | 0.84× |
| R-Stitch | 0.001 | 40.90 | 160.10 | 1.14× | 68.85 | 7.13 | 1.06× | 77.40 | 16.65 | 1.51× | 62.38 | 61.29 | 1.17× |
| R-Stitch | 0.005 | 36.20 | 142.03 | 1.29× | 62.40 | 5.83 | 1.29× | 70.10 | 8.06 | 3.11× | 56.23 | 51.97 | 1.38× |
| R-Stitch | 0.01 | 37.78 | 131.43 | 1.39× | 56.10 | 4.30 | 1.75× | 73.20 | 5.86 | 4.28× | 55.69 | 47.20 | 1.52× |

## A.8 PERFORMANCE ON CODE GENERATION BENCHMARKS.

Table 4 reports accuracy, latency, and relative speedup on programming benchmarks including LiveCodeBench (Jain et al., 2025), MBPP (Austin et al., 2021), and HumanEval (Chen et al., 2021). Compared to math reasoning, acceleration on code generation is more modest: when targeting accuracy comparable to the full LLM, the achievable latency reduction is limited. We attribute this to the characteristics of the specific SLM used in our study, which, while effective on math reasoning tasks, produces less concise and less reliable traces for code. This constrains the gains realizable from hybrid decoding on code benchmarks. Nevertheless, R-Stitch still enables a flexible trade-off between efficiency and accuracy across model scales. By adjusting the entropy threshold, one can smoothly interpolate between full LLM accuracy and substantially lower latency, yielding deployment options adapted to different computational budgets and latency requirements. This shows that, even when the paired SLM is less competitive, token-level hybrid decoding offers a principled and flexible mechanism to balance quality and efficiency, ensuring that practitioners can extract most of the benefits of large models at significantly reduced cost.

## A.9 COMPARISON WITH SAME-FAMILY SPECULATIVE DECODING

We report results where speculative decoding is evaluated using both the model pairs in the manuscript and the same-family DeepSeek-R1–Distill-Qwen draft–target pairs with higher consistency (marked with *). All results in Table 5 show accuracy (%) and latency (s) under an 8k budget. We highlight in **bold** the best acceleration achieved without any accuracy drop. While same-family pairing improves speculative decoding by increasing draft–target agreement, its acceleration remains limited because strict token-level consistency is still required. R-Stitch achieves comparable or better accuracy with substantially lower latency, as it does not rely on distributional alignment and can exploit concise community SLMs across model families without retraining. Extended comparisons have been added to Appendix A.9 in the revised manuscript.

Table 5: Results of speculative decoding with same-family and manuscript model pairs.

| Method | AIME | | AMC | | MATH | | Minerva | |
|---|---|---|---|---|---|---|---|---|
| | Acc ↑ | Lat. ↓ | Acc ↑ | Lat. ↓ | Acc ↑ | Lat. ↓ | Acc ↑ | Lat. ↓ |
| LLM 7B | 33.33 | 86.63 | 66.27 | 63.15 | 86.00 | 38.34 | 31.62 | 41.79 |
| SpecDec 7B | 36.67 | 201.23 | 69.88 | 95.42 | 87.00 | 48.71 | 34.19 | 56.59 |
| R-Stitch 7B | **40.00** | **62.03** | **69.88** | **34.89** | **87.00** | **16.61** | **33.09** | **18.98** |
| SpecDec 7B* | 33.33 | 68.65 | 71.08 | 58.53 | 89.20 | 36.86 | 34.19 | 43.33 |
| R-Stitch 7B* | 36.67 | 69.73 | 68.67 | 61.16 | 89.20 | 40.55 | 36.76 | 41.15 |
| LLM 14B | 43.33 | 153.20 | 68.67 | 101.76 | 86.00 | 41.66 | 35.29 | 48.02 |
| SpecDec 14B | 50.00 | 139.10 | 68.67 | 95.59 | 82.80 | 45.88 | 33.46 | 53.02 |
| R-Stitch 14B | **43.33** | **87.61** | **69.88** | **41.82** | **85.20** | **20.53** | **35.66** | **22.16** |
| SpecDec 14B* | 50.00 | 127.59 | 74.70 | 71.39 | 83.80 | 37.02 | 33.46 | 40.95 |
| R-Stitch 14B* | 43.33 | 112.23 | 74.70 | 73.24 | 85.20 | 35.41 | 35.66 | 38.89 |

## A.10 BATCH SIZE ANALYSIS

We report here the effect of batch size on R-Stitch latency. Because R-Stitch performs token-level switching between the SLM and LLM, the acceleration ratio naturally decreases as batch size grows; this behavior is shared by speculative-decoding–style methods, where alternating model invocations introduce synchronization points and reduce GPU utilization.

We evaluate R-Stitch and speculative decoding on the MATH dataset using the DeepSeek-R1–Distill-Qwen-7B pair within vLLM, with $\tau = 0.02$. The full LLM reaches 86.00% accuracy, and both acceleration methods obtain 87.00% accuracy. Table 6 reports average per-sample latency under each batch size. Although acceleration benefits diminish at larger batches, R-Stitch consistently achieves lower latency than speculative decoding, owing to its ability to exploit concise SLM reasoning traces beyond strict token-level consistency.

Table 6: Latency (s) of the LLM, speculative decoding, and R-Stitch across batch sizes on MATH.

| Method / Batch | 1 | 2 | 4 | 8 | 16 | 32 | 64 | 128 |
|---|---|---|---|---|---|---|---|---|
| LLM | 38.34 | 31.58 | 22.07 | 16.11 | 11.08 | 8.42 | 6.52 | 5.97 |
| SpecDec | 36.86 | 31.61 | 24.29 | 22.04 | 19.20 | 17.88 | 15.32 | 14.01 |
| R-Stitch | 16.61 | 20.48 | 17.98 | 14.98 | 12.02 | 10.61 | 7.43 | 6.55 |

## A.11 MEMORY OVERHEAD OF DUAL KV CACHES

R-Stitch, like all speculative-decoding–style approaches, maintains separate KV caches for the SLM and LLM. To quantify the actual overhead, we measure model weights and KV footprints in `vLLM` under BF16 precision. Table 7 reports per-token KV size, model weights, and total memory usage at 8k and 16k context lengths.

Table 7: Memory usage of SLM/LLM models and their KV caches under BF16.

| Model size | Per-token KV (MB) | Weights (MB) | Total @8k (MB) | Total @16k (MB) |
|---|---|---|---|---|
| 1.5B | 0.028 | 3036 | 3265 | 3494 |
| 7B | 0.055 | 15640 | 16091 | 16542 |
| 14B | 0.189 | 29275 | 30824 | 32374 |
| 32B | 0.253 | 63890 | 65958 | 68026 |

When paired with a 7B model under a 16k context, the 1.5B SLM introduces only a 17.44% memory increase, which further decreases to 9.74% for a 14B model and 4.89% for a 32B model. These

results show that the additional memory required by R-Stitch remains modest relative to the LLM footprint, while enabling substantial end-to-end latency reductions.

## A.12 CROSS-DOMAIN EVALUATION

We evaluate R-Stitch ($\tau = 0.02$) and R-Stitch+ beyond mathematical reasoning on four QA and reasoning benchmarks: GPQA-D (Rein et al., 2024), ZebraLogicBench (Lin et al.), CRUXEval (Gu et al.), and MMLU-Redux (Gema et al., 2025). Accuracy ($\uparrow$) and latency ($\downarrow$) are reported. For R-Stitch+, the router is trained only on math datasets and directly applied to these domains without any retraining.

Table 8: Cross-domain evaluation on reasoning and QA benchmarks.

| Method | GPQA-D | | ZebraLogicBench | | CRUXEval | | MMLU-Redux | |
|---|---|---|---|---|---|---|---|---|
| | Acc ↑ | Lat. ↓ | Acc ↑ | Lat. ↓ | Acc ↑ | Lat. ↓ | Acc ↑ | Lat. ↓ |
| LLM-7B | 23.74 | 171.76 | 21.09 | 91.85 | 70.25 | 14.54 | 74.64 | 20.10 |
| SpecDec | 24.24 | 152.23 | 20.22 | 72.03 | 70.50 | 14.78 | 74.04 | 19.37 |
| R-Stitch | 35.86 | 44.57 | 20.78 | 48.23 | 87.29 | 18.23 | 74.04 | 8.12 |
| R-Stitch+ | 34.34 | 39.02 | 20.82 | 45.12 | 69.88 | 9.15 | 74.11 | 8.52 |

R-Stitch maintains accuracy comparable to the LLM across all tasks while providing substantial latency reductions. Moreover, R-Stitch+ consistently delivers slightly improved accuracy despite being trained only on math data, showing that its learned policy transfers well across tasks without retraining.

## A.13 ARCHITECTURE AND OVERHEAD OF ROUTER

The router is explicitly designed to be lightweight, and its computational and memory overhead is negligible. It is a small feed-forward network operating on SLM hidden states, composed of 6 FFN blocks with GELU and residual connections, followed by normalization and a two-head linear layer. The input matches the SLM hidden size (3584 for the 1.5B model), and the total parameter count is 0.17M. On an NVIDIA A100, the router adds 1.05 ms per invocation. At $\tau = 0.02$ on AIME with the 7B pair, it is invoked 350.38 times on average, giving 368 ms total overhead. The full inference time for the same setup is 62.03 s, so the router contributes under 0.6% of end-to-end latency, demonstrating that its deployability costs are minimal.

## A.14 COMPUTATIONAL OVERHEAD OF ENTROPY-BASED ROUTING

The cost of entropy evaluation is negligible: entropy is obtained directly from the SLM logits and adds only 0.028 ms per token in our implementation. In the 7B–1.5B AIME setup ($\tau = 0.02$), decoding 4530.60 tokens takes 62.03 seconds, while entropy computation accounts for just 127 ms (approximately 0.2%), indicating no meaningful runtime burden.

## A.15 COMPARISON WITH REWARD-GUIDED SPECULATIVE DECODING

We compare R-Stitch with Reward-Guided Speculative Decoding (RSD) (Liao et al.), which performs step-level switching based on an external process reward model (PRM). Since PRMs are trained for specific target models and transfer poorly across different model pairs, RSD's routing decisions become unstable when the draft–target model combination changes. In contrast, R-Stitch performs fine-grained token-level routing without any external supervision, providing more stable efficiency

Table 9: Comparison between RSD and R-Stitch under an 8k decoding budget.

| Method | AIME | | AMC | | MATH | | Minerva | |
|---|---|---|---|---|---|---|---|---|
| | Acc ↑ | Lat. ↓ | Acc ↑ | Lat. ↓ | Acc ↑ | Lat. ↓ | Acc ↑ | Lat. ↓ |
| LLM-7B | 33.33 | 86.63 | 66.27 | 63.15 | 86.00 | 38.34 | 31.62 | 41.79 |
| RSD-7B | 30.00 | 69.73 | 66.27 | 54.12 | 81.60 | 25.12 | 33.80 | 54.37 |
| R-Stitch-7B | 40.00 | 62.03 | 69.88 | 34.89 | 87.00 | 16.61 | 33.09 | 18.98 |

gains across diverse model pairs. Table 9 reports accuracy (%) and latency (s) under an 8k decoding budget.

### A.16    MORE PER-SAMPLE COMPARISONS WITH SPECULATIVE DECODING

Figures 7 and 8 present additional per-sample visualizations of latency speedup and token reduction when applying R-Stitch$^+$ and Speculative Decoding on the LLM-7B model across more datasets beyond those reported in the main text. The supplementary results confirm the same pattern: R-Stitch$^+$ consistently accelerates the majority of samples, whereas Speculative Decoding provides gains only on a minority and often leads to slowdowns due to consistency issues.

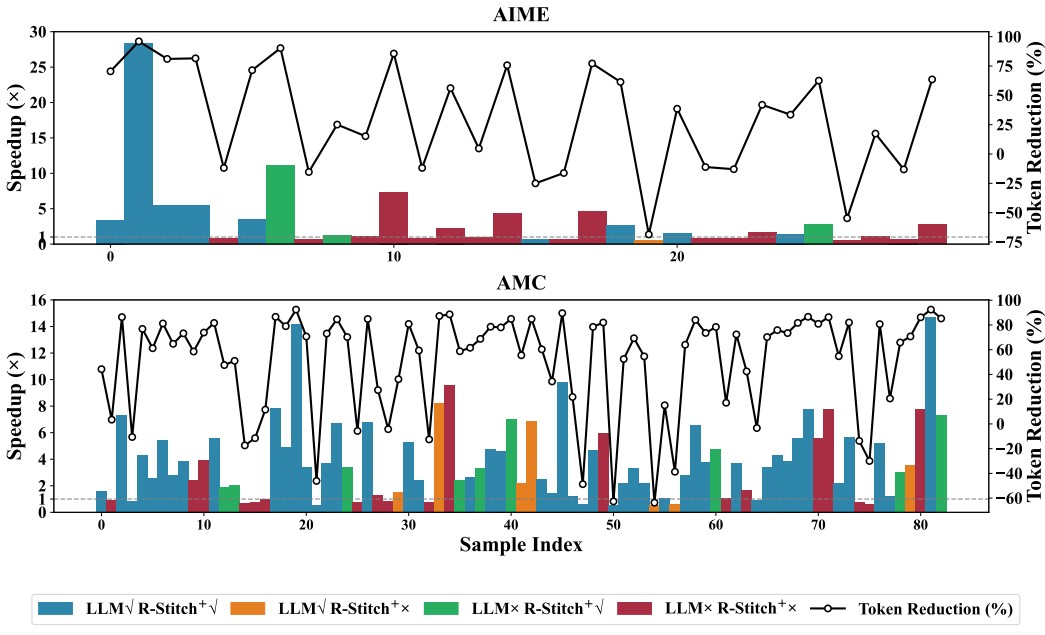

Figure 7: **Per-sample visualization of R-Stitch$^+$ (LLM-7B).** Each bar shows the latency speedup of one sample relative to the baseline LLM. Bar colors encode correctness outcomes of the baseline and R-Stitch$^+$. The dashed horizontal line at 1 indicates no speedup. The black solid curve with hollow-circle markers represents token reduction percentages per sample.

### A.17    ADDITIONAL RESULTS ON TOKEN USAGE

In this section, we provide detailed tables for each dataset and benchmark under different decoding budgets as shown in Table 10 to Table 19. Beyond the main results on accuracy, latency, and relative speedup, these tables additionally report the number of tokens consumed by the LLM and the SLM, as well as the total token usage. We also compute the percentage of token reduction compared to full LLM decoding. These results offer a clearer view of how R-Stitch reduces the reliance on expensive LLM tokens across mathematical reasoning and code generation tasks, while maintaining competitive accuracy and achieving substantial efficiency improvements.

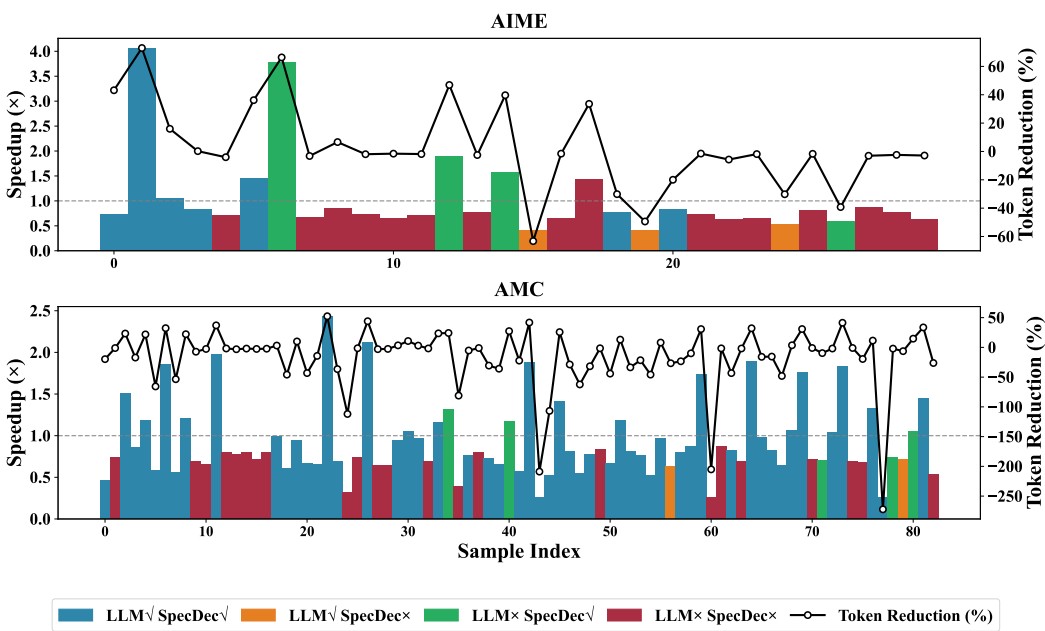

Figure 8: **Per-sample visualization of Speculative Decoding (LLM-7B).** Each bar shows the latency speedup of one sample relative to the baseline LLM. Bar colors encode correctness outcomes of the baseline and Speculative Decoding. The dashed horizontal line at 1 indicates no speedup. The black solid curve with hollow-circle markers represents token reduction percentages per sample.

Table 10: Comparison on **AIME** with decoding budget = 8k tokens.

| Method | $\tau$ | Acc | Lat. | Spd. | LLM Tok. | SLM Tok. | Total Tok. | Reduction (%) |
|--------|--------|-----|------|------|----------|----------|------------|---------------|
| SLM | – | 10.00 | 5.91 | – | – | – | 478.27 | – |
| LLM-7B | – | 33.33 | 86.63 | 1.00× | – | – | 6699.63 | – |
| SpecDec | – | 36.67 | 201.23 | 0.43× | – | – | 6922.83 | +3.33% |
| R-Stitch | 0.001 | 36.67 | 89.86 | 0.96× | 3970.03 | 2916.57 | 6886.60 | +2.79% |
| R-Stitch | 0.02 | 40.00 | 62.03 | 1.40× | 2607.24 | 1923.36 | 4530.60 | -32.38% |
| R-Stitch | 0.03 | 30.00 | 42.06 | 2.06× | 1052.17 | 1914.37 | 2966.54 | -55.72% |
| LLM-14B | – | 43.33 | 153.20 | 1.00× | – | – | 6243.13 | – |
| SpecDec | – | 50.00 | 139.10 | 1.10× | – | – | 6192.73 | -0.81% |
| R-Stitch | 0.001 | 50.00 | 129.88 | 1.18× | 3080.07 | 2498.33 | 5578.40 | -10.65% |
| R-Stitch | 0.02 | 43.33 | 87.61 | 1.75× | 1693.90 | 2386.07 | 4079.97 | -34.65% |
| R-Stitch | 0.03 | 43.33 | 61.59 | 2.49× | 1030.93 | 2040.03 | 3070.96 | -50.81% |
| LLM-32B | – | 43.33 | 354.85 | 1.00× | – | – | 7334.00 | – |
| SpecDec | – | 40.00 | 270.25 | 1.31× | – | – | 7540.83 | +2.82% |
| R-Stitch | 0.001 | 50.00 | 261.86 | 1.36× | 4180.43 | 2613.80 | 6794.23 | -7.36% |
| R-Stitch | 0.02 | 50.00 | 184.97 | 1.92× | 2486.10 | 3085.17 | 5571.27 | -24.04% |
| R-Stitch | 0.03 | 40.00 | 178.19 | 1.99× | 2284.67 | 3352.27 | 5636.94 | -23.14% |

## A.18 QUALITATIVE CASE STUDIES

To provide a more intuitive understanding of how R-Stitch leverages the complementary strengths of the SLM and LLM, we visualize several representative examples. For each selected problem, we compare the responses of the SLM, the LLM, speculative decoding, and our method R-Stitch with $\tau = 0.02$. For long outputs, we omit intermediate tokens for readability.

As shown in Figure 9 to Figure 12, the SLM along generates very concise answers with extremely low latency but often produces incorrect solutions. The LLM eventually produces correct answers but requires thousands of tokens and much higher latency. Speculative decoding sometimes suffers from frequent token rejections under low model agreement, resulting in more decoding steps than

Table 11: Comparison on **AIME** with decoding budget = 16k tokens.

| Method | $\tau$ | Acc | Lat. | Spd. | LLM Tok. | SLM Tok. | Total Tok. | Reduction (%) |
|---|---|---|---|---|---|---|---|---|
| SLM | – | 10.00 | 5.91 | – | – | – | 478.27 | – |
| LLM-7B | – | 40.00 | 195.77 | 1.00× | – | – | 11739.10 | – |
| SpecDec | – | 50.00 | 258.54 | 0.76× | – | – | 12478.27 | +6.30% |
| R-Stitch | 0.001 | 46.67 | 192.22 | 1.02× | 5283.83 | 3898.57 | 9182.40 | -21.78% |
| R-Stitch | 0.02 | 50.00 | 110.16 | 1.78× | 2325.67 | 3402.33 | 5728.00 | -51.21% |
| R-Stitch | 0.03 | 36.67 | 41.51 | 4.72× | 920.10 | 1753.27 | 2673.37 | -77.23% |
| R-Stitch+ | – | 50.00 | 86.03 | 2.28× | 1911.24 | 2345.13 | 4256.37 | -63.74% |
| LLM-14B | – | 50.00 | 246.95 | 1.00× | – | – | 9130.13 | – |
| SpecDec | – | 50.00 | 275.64 | 0.90× | – | – | 6312.04 | -30.87% |
| R-Stitch | 0.001 | 56.67 | 217.06 | 1.14× | 4476.53 | 3550.63 | 8027.16 | -12.08% |
| R-Stitch | 0.02 | 43.33 | 99.12 | 2.49× | 1746.50 | 2654.30 | 4400.80 | -51.80% |
| R-Stitch | 0.03 | 40.00 | 54.89 | 4.50× | 983.83 | 1813.63 | 2797.46 | -69.36% |
| R-Stitch+ | – | 50.00 | 132.98 | 1.86× | 2784.67 | 2367.77 | 5152.44 | -43.57% |
| LLM-32B | – | 56.67 | 591.67 | 1.00× | – | – | 11602.67 | – |
| SpecDec | – | 70.00 | 526.03 | 1.12× | – | – | 10879.17 | -6.24% |
| R-Stitch | 0.001 | 70.00 | 488.87 | 1.21× | 6963.43 | 3908.77 | 10872.20 | -6.30% |
| R-Stitch | 0.02 | 50.00 | 333.17 | 1.78× | 4119.00 | 4431.53 | 8550.53 | -26.31% |
| R-Stitch | 0.03 | 53.33 | 276.03 | 2.14× | 3302.57 | 4098.20 | 7400.77 | -36.21% |

Table 12: Comparison on **AMC** with decoding budget = 8k tokens.

| Method | $\tau$ | Acc | Lat. | Spd. | LLM Tok. | SLM Tok. | Total Tok. | Reduction (%) |
|---|---|---|---|---|---|---|---|---|
| SLM | – | 50.60 | 5.37 | – | – | – | 437.29 | – |
| LLM-7B | – | 66.27 | 63.15 | 1.00× | – | – | 4796.87 | – |
| SpecDec | – | 69.88 | 95.42 | 0.66× | – | – | 4922.23 | +2.61% |
| R-Stitch | 0.001 | 77.11 | 58.72 | 1.08× | 1868.92 | 1792.31 | 3661.23 | -23.67% |
| R-Stitch | 0.02 | 69.88 | 34.89 | 1.81× | 1365.77 | 1208.35 | 2574.12 | -46.34% |
| R-Stitch | 0.03 | 69.88 | 24.55 | 2.57× | 484.27 | 1165.06 | 1649.33 | -65.62% |
| R-Stitch+ | – | 68.67 | 21.08 | 3.00× | 468.72 | 987.24 | 1455.96 | -69.65% |
| LLM-14B | – | 68.67 | 101.76 | 1.00× | – | – | 4199.61 | – |
| SpecDec | – | 68.67 | 95.59 | 1.06× | – | – | 4495.41 | +7.04% |
| R-Stitch | 0.001 | 69.88 | 79.54 | 1.28× | 1799.34 | 1803.77 | 3603.11 | -14.20% |
| R-Stitch | 0.02 | 69.88 | 41.82 | 2.43× | 756.01 | 1366.87 | 2122.88 | -49.45% |
| R-Stitch | 0.03 | 68.67 | 26.43 | 3.85× | 453.52 | 984.59 | 1438.11 | -65.76% |
| R-Stitch+ | – | 73.49 | 49.91 | 2.04× | 1137.86 | 1135.87 | 2273.73 | -45.86% |
| LLM-32B | – | 60.24 | 292.78 | 1.00× | – | – | 6081.42 | – |
| SpecDec | – | 59.04 | 209.72 | 1.40× | – | – | 6102.70 | +0.35% |
| R-Stitch | 0.001 | 68.67 | 209.88 | 1.39× | 3094.76 | 2131.43 | 5226.19 | -14.06% |
| R-Stitch | 0.02 | 68.67 | 118.26 | 2.48× | 1567.72 | 2175.42 | 3743.14 | -38.45% |
| R-Stitch | 0.03 | 69.88 | 86.88 | 3.37× | 1100.34 | 1911.72 | 3012.06 | -50.47% |

vanilla LLM decoding and hence longer latency. In contrast, R-Stitch effectively combines the SLM's brevity with the LLM's accuracy: it selectively accepts confident SLM tokens while routing uncertain ones to the LLM, producing shorter but still correct solutions. This substantially reduces the number of generated tokens and yields significant acceleration.

Table 13: Comparison on **AMC** with decoding budget = 16k tokens.

| Method | $\tau$ | Acc | Lat. | Spd. | LLM Tok. | SLM Tok. | Total Tok. | Reduction (%) |
|---|---|---|---|---|---|---|---|---|
| SLM | – | 50.60 | 5.37 | – | – | – | 437.29 | – |
| LLM-7B | – | 71.08 | 116.41 | 1.00× | – | – | 7225.24 | – |
| SpecDec | – | 80.72 | 132.45 | 0.88× | – | – | 9211.29 | +27.49% |
| R-Stitch | 0.001 | 80.72 | 97.26 | 1.20× | 2902.64 | 2367.42 | 5270.06 | -27.06% |
| R-Stitch | 0.02 | 67.47 | 54.17 | 2.15× | 1212.94 | 1830.86 | 3043.80 | -57.87% |
| R-Stitch | 0.03 | 66.27 | 38.07 | 3.06× | 725.36 | 1505.22 | 2230.58 | -69.13% |
| R-Stitch$^+$ | – | 77.11 | 56.03 | 2.08× | 1162.74 | 1632.52 | 2795.26 | -61.31% |
| LLM-14B | – | 86.75 | 146.51 | 1.00× | – | – | 5619.14 | – |
| SpecDec | – | 83.13 | 138.53 | 1.06× | – | – | 4310.51 | -23.29% |
| R-Stitch | 0.001 | 79.52 | 90.72 | 1.61× | 1968.23 | 1906.05 | 3874.28 | -31.05% |
| R-Stitch | 0.02 | 66.27 | 51.29 | 2.86× | 900.52 | 1524.18 | 2424.70 | -56.85% |
| R-Stitch | 0.03 | 63.86 | 28.38 | 5.16× | 460.84 | 1062.13 | 1522.97 | -72.90% |
| R-Stitch$^+$ | – | 78.31 | 68.50 | 2.14× | 1476.96 | 1450.69 | 2927.65 | -47.90% |
| LLM-32B | – | 87.95 | 419.94 | 1.00× | – | – | 8384.20 | – |
| SpecDec | – | 87.95 | 326.73 | 1.29× | – | – | 8217.63 | -1.99% |
| R-Stitch | 0.001 | 90.36 | 288.92 | 1.45× | 4251.51 | 2716.18 | 6967.69 | -16.89% |
| R-Stitch | 0.02 | 81.93 | 168.19 | 2.50× | 2081.64 | 2629.07 | 4710.71 | -43.81% |
| R-Stitch | 0.03 | 73.49 | 143.53 | 2.93× | 1639.04 | 2661.71 | 4300.75 | -48.70% |

Table 14: Comparison on **Minerva** with decoding budget = 8k tokens.

| Method | $\tau$ | Acc | Lat. | Spd. | LLM Tok. | SLM Tok. | Total Tok. | Reduction (%) |
|---|---|---|---|---|---|---|---|---|
| SLM | – | 25.37 | 5.03 | – | – | – | 408.08 | – |
| LLM-7B | – | 31.62 | 41.79 | 1.00× | – | – | 2966.47 | – |
| SpecDec | – | 34.19 | 56.59 | 0.74× | – | – | 3293.91 | +11.04% |
| R-Stitch | 0.001 | 34.19 | 38.73 | 1.08× | 1320.15 | 1141.49 | 2461.64 | -17.02% |
| R-Stitch | 0.02 | 33.09 | 18.98 | 2.20× | 455.23 | 820.51 | 1275.74 | -56.99% |
| R-Stitch | 0.03 | 33.09 | 17.72 | 2.36× | 375.65 | 803.61 | 1179.26 | -60.25% |
| R-Stitch$^+$ | – | 35.29 | 15.33 | 2.73× | 398.68 | 756.12 | 1154.80 | -61.07% |
| LLM-14B | – | 35.29 | 48.02 | 1.00× | – | – | 3348.93 | – |
| SpecDec | – | 34.93 | 53.02 | 0.91× | – | – | 3216.78 | -3.95% |
| R-Stitch | 0.001 | 35.66 | 40.54 | 1.18× | 1014.94 | 925.07 | 1940.01 | -42.07% |
| R-Stitch | 0.02 | 35.66 | 22.16 | 2.17× | 409.41 | 776.69 | 1186.10 | -64.58% |
| R-Stitch | 0.03 | 34.93 | 17.59 | 2.73× | 299.85 | 674.68 | 974.53 | -70.90% |
| R-Stitch$^+$ | – | 35.66 | 26.66 | 1.80× | 775.71 | 763.89 | 1539.60 | -54.03% |
| LLM-32B | – | 41.18 | 231.10 | 1.00× | – | – | 4720.20 | – |
| SpecDec | – | 41.91 | 182.28 | 1.27× | – | – | 4724.27 | +0.09% |
| R-Stitch | 0.001 | 42.65 | 141.24 | 1.64× | 2309.25 | 1529.28 | 3838.53 | -18.68% |
| R-Stitch | 0.02 | 36.76 | 59.58 | 3.88× | 831.41 | 1176.50 | 2007.91 | -57.46% |
| R-Stitch | 0.03 | 34.56 | 43.41 | 5.32× | 578.15 | 1002.49 | 1580.64 | -66.51% |

Table 15: Comparison on **Minerva** with decoding budget = 16k tokens.

| Method | $\tau$ | Acc | Lat. | Spd. | LLM Tok. | SLM Tok. | Total Tok. | Reduction (%) |
|---|---|---|---|---|---|---|---|---|
| SLM | – | 25.37 | 5.03 | – | – | – | 408.08 | – |
| LLM-7B | – | 34.93 | 54.43 | 1.00× | – | – | 3616.44 | – |
| SpecDec | – | 35.66 | 115.76 | 0.47× | – | – | 5987.29 | +65.56% |
| R-Stitch | 0.001 | 35.66 | 45.53 | 1.20× | 1482.54 | 1272.18 | 2754.72 | -23.83% |
| R-Stitch | 0.02 | 33.09 | 18.79 | 2.90× | 443.68 | 810.42 | 1254.10 | -65.32% |
| R-Stitch | 0.03 | 35.29 | 13.50 | 4.03× | 295.17 | 663.07 | 958.24 | -73.50% |
| R-Stitch$^+$ | – | 35.29 | 13.68 | 3.98× | 283.08 | 667.68 | 950.76 | -73.71% |
| LLM-14B | – | 39.34 | 68.99 | 1.00× | – | – | 2801.30 | – |
| SpecDec | – | 39.34 | 55.40 | 1.25× | – | – | 3521.18 | +25.70% |
| R-Stitch | 0.001 | 37.87 | 40.46 | 1.71× | 990.31 | 924.97 | 1915.28 | -31.63% |
| R-Stitch | 0.02 | 31.99 | 22.74 | 3.03× | 416.56 | 784.71 | 1201.27 | -57.12% |
| R-Stitch | 0.03 | 33.82 | 16.14 | 4.27× | 263.77 | 645.72 | 909.49 | -67.53% |
| R-Stitch$^+$ | – | 37.13 | 35.11 | 1.96× | 810.15 | 810.38 | 1620.53 | -42.15% |
| LLM-32B | – | 46.69 | 348.48 | 1.00× | – | – | 5553.52 | – |
| SpecDec | – | 44.23 | 273.07 | 1.28× | – | – | 5982.18 | +7.72% |
| R-Stitch | 0.001 | 41.18 | 126.62 | 2.75× | 2013.80 | 1605.40 | 3619.20 | -34.83% |
| R-Stitch | 0.02 | 40.07 | 73.32 | 4.75× | 900.55 | 1923.00 | 2823.55 | -49.16% |
| R-Stitch | 0.03 | 36.40 | 43.58 | 8.00× | 302.19 | 626.28 | 928.47 | -83.28% |

Table 16: Comparison on **MATH** with decoding budget = 8k tokens.

| Method | $\tau$ | Acc | Lat. | Spd. | LLM Tok. | SLM Tok. | Total Tok. | Reduction (%) |
|---|---|---|---|---|---|---|---|---|
| SLM | – | 73.60 | 4.56 | – | – | – | 370.67 | – |
| LLM-7B | – | 86.00 | 38.34 | 1.00× | – | – | 2753.01 | – |
| SpecDec | – | 87.00 | 48.71 | 0.79× | – | – | 3048.78 | +10.74% |
| R-Stitch | 0.001 | 89.40 | 32.94 | 1.16× | 1022.38 | 1101.67 | 2124.05 | -22.85% |
| R-Stitch | 0.02 | 87.00 | 16.61 | 2.31× | 344.28 | 770.52 | 1114.80 | -59.51% |
| R-Stitch | 0.03 | 85.60 | 15.15 | 2.53× | 280.26 | 748.21 | 1028.47 | -62.64% |
| R-Stitch$^+$ | – | 86.60 | 15.68 | 2.45× | 503.17 | 598.88 | 1102.05 | -59.97% |
| LLM-14B | – | 86.00 | 41.66 | 1.00× | – | – | 2934.52 | – |
| SpecDec | – | 82.80 | 45.88 | 0.91× | – | – | 2443.78 | -16.72% |
| R-Stitch | 0.001 | 89.00 | 37.03 | 1.13× | 867.72 | 937.65 | 1805.37 | -38.48% |
| R-Stitch | 0.02 | 85.20 | 20.53 | 2.03× | 349.82 | 749.75 | 1099.57 | -62.53% |
| R-Stitch | 0.03 | 83.40 | 14.77 | 2.82× | 215.10 | 645.26 | 860.36 | -70.68% |
| R-Stitch$^+$ | – | 88.20 | 29.08 | 1.43× | 635.27 | 764.58 | 1399.85 | -52.30% |
| LLM-32B | – | 87.20 | 178.38 | 1.00× | – | – | 3054.16 | – |
| SpecDec | – | 88.60 | 136.73 | 1.30× | – | – | 3052.70 | -0.05% |
| R-Stitch | 0.001 | 91.20 | 95.49 | 1.87× | 1507.99 | 1322.78 | 2830.77 | -7.31% |
| R-Stitch | 0.02 | 90.60 | 80.21 | 2.22× | 1087.60 | 1209.95 | 2297.55 | -24.77% |
| R-Stitch | 0.03 | 87.00 | 32.21 | 5.54× | 387.09 | 890.72 | 1277.81 | -58.16% |

Table 17: Comparison on **MATH** with decoding budget = 16k tokens.

| Method | $\tau$ | Acc | Lat. | Spd. | LLM Tok. | SLM Tok. | Total Tok. | Reduction (%) |
|---|---|---|---|---|---|---|---|---|
| SLM | – | 73.60 | 4.56 | – | – | – | 370.67 | – |
| LLM-7B | – | 90.80 | 50.07 | 1.00× | – | – | 3381.51 | – |
| SpecDec | – | 91.20 | 44.47 | 1.13× | – | – | 2645.27 | -21.77% |
| R-Stitch | 0.001 | 91.00 | 38.34 | 1.31× | 1136.11 | 1151.15 | 2287.26 | -32.36% |
| R-Stitch | 0.02 | 88.60 | 22.56 | 2.22× | 451.49 | 908.54 | 1360.03 | -59.78% |
| R-Stitch | 0.03 | 83.20 | 15.62 | 3.21× | 275.91 | 729.55 | 1005.46 | -70.27% |
| R-Stitch$^+$ | – | 88.60 | 16.02 | 3.13× | 278.17 | 732.16 | 1010.33 | -70.12% |
| LLM-14B | – | 88.60 | 66.54 | 1.00× | – | – | 2698.10 | – |
| SpecDec | – | 87.40 | 67.18 | 0.99× | – | – | 2681.58 | -0.61% |
| R-Stitch | 0.001 | 89.40 | 39.82 | 1.67× | 886.11 | 958.29 | 1844.40 | -31.64% |
| R-Stitch | 0.02 | 87.80 | 22.62 | 2.94× | 365.23 | 793.84 | 1159.07 | -57.04% |
| R-Stitch | 0.03 | 84.80 | 17.37 | 3.83× | 237.71 | 687.69 | 925.40 | -65.70% |
| R-Stitch$^+$ | – | 89.60 | 33.70 | 1.97× | 719.47 | 811.61 | 1531.08 | -43.25% |
| LLM-32B | – | 94.00 | 249.92 | 1.00× | – | – | 4304.52 | – |
| SpecDec | – | 93.20 | 187.79 | 1.33× | – | – | 4023.25 | -6.53% |
| R-Stitch | 0.001 | 94.00 | 172.08 | 1.45× | 952.53 | 905.83 | 1858.36 | -56.83% |
| R-Stitch | 0.02 | 90.60 | 80.21 | 3.12× | 1087.60 | 1209.95 | 2297.55 | -46.62% |
| R-Stitch | 0.03 | 87.80 | 38.32 | 6.52× | 148.40 | 467.67 | 616.07 | -85.69% |

Table 18: Comparison on **OlympiadBench** with decoding budget = 8k tokens.

| Method | $\tau$ | Acc | Lat. | Spd. | LLM Tok. | SLM Tok. | Total Tok. | Reduction (%) |
|---|---|---|---|---|---|---|---|---|
| SLM | – | 36.89 | 5.42 | – | – | – | 441.54 | – |
| LLM-7B | – | 51.85 | 117.31 | 1.00× | – | – | 5109.47 | – |
| SpecDec | – | 51.85 | 134.23 | 0.87× | – | – | 5292.34 | +3.58% |
| R-Stitch | 0.001 | 55.26 | 105.29 | 1.11× | 1551.23 | 1216.72 | 2767.95 | -45.83% |
| R-Stitch | 0.02 | 51.85 | 70.43 | 1.67× | 487.93 | 810.37 | 1298.30 | -74.59% |
| R-Stitch | 0.03 | 48.59 | 51.43 | 2.28× | 280.62 | 647.15 | 927.77 | -81.84% |
| R-Stitch$^+$ | – | 52.00 | 45.02 | 2.61× | 374.52 | 679.23 | 1053.75 | -79.38% |
| LLM-14B | – | 54.07 | 190.89 | 1.00× | – | – | 4623.48 | – |
| SpecDec | – | 54.22 | 180.03 | 1.06× | – | – | 5320.88 | +15.08% |
| R-Stitch | 0.001 | 54.22 | 165.07 | 1.16× | 1285.81 | 938.74 | 2224.55 | -51.89% |
| R-Stitch | 0.02 | 52.44 | 96.09 | 1.99× | 375.46 | 668.54 | 1044.00 | -77.42% |
| R-Stitch | 0.03 | 48.44 | 52.36 | 3.65× | 214.74 | 556.72 | 771.46 | -83.31% |
| R-Stitch$^+$ | – | 56.30 | 108.67 | 1.76× | 906.33 | 809.67 | 1716.00 | -62.89% |
| LLM-32B | – | 50.67 | 379.52 | 1.00× | – | – | 5988.08 | – |
| SpecDec | – | 50.37 | 351.27 | 1.08× | – | – | 4564.46 | -23.77% |
| R-Stitch | 0.001 | 50.67 | 341.67 | 1.11× | 2441.57 | 1432.92 | 3874.49 | -35.30% |
| R-Stitch | 0.02 | 53.78 | 226.71 | 1.67× | 488.88 | 781.20 | 1270.08 | -78.79% |
| R-Stitch | 0.03 | 52.15 | 157.47 | 2.41× | 515.87 | 968.86 | 1484.73 | -75.21% |

Table 19: Comparison on **OlympiadBench** with decoding budget = 16k tokens.

| Method | $\tau$ | Acc | Lat. | Spd. | LLM Tok. | SLM Tok. | Total Tok. | Reduction (%) |
|---|---|---|---|---|---|---|---|---|
| SLM | – | 36.89 | 5.42 | – | – | – | 441.54 | – |
| LLM-7B | – | 58.67 | 208.77 | 1.00× | – | – | 6839.96 | – |
| SpecDec | – | 60.15 | 267.03 | 0.78× | – | – | 7208.65 | +5.39% |
| R-Stitch | 0.001 | 60.15 | 201.13 | 1.04× | 1817.81 | 1327.43 | 3145.24 | -54.02% |
| R-Stitch | 0.02 | 58.67 | 109.98 | 1.90× | 504.00 | 840.55 | 1344.55 | -80.34% |
| R-Stitch | 0.03 | 54.22 | 65.65 | 3.18× | 288.84 | 664.50 | 953.34 | -86.06% |
| R-Stitch$^+$ | – | 59.11 | 95.31 | 2.19× | 498.17 | 814.36 | 1312.53 | -80.81% |
| LLM-14B | – | 60.00 | 316.06 | 1.00× | – | – | 6024.14 | – |
| SpecDec | – | 60.00 | 360.14 | 0.88× | – | – | 6143.46 | +1.98% |
| R-Stitch | 0.001 | 61.04 | 69.28 | 4.56× | 1516.14 | 1225.07 | 2741.21 | -54.50% |
| R-Stitch | 0.02 | 54.22 | 108.61 | 2.91× | 449.36 | 771.67 | 1221.03 | -79.73% |
| R-Stitch | 0.03 | 48.59 | 68.63 | 4.61× | 257.66 | 623.88 | 881.54 | -85.37% |
| R-Stitch$^+$ | – | 59.11 | 121.39 | 2.60× | 982.06 | 846.84 | 1828.90 | -69.64% |
| LLM-32B | – | 67.70 | 722.77 | 1.00× | – | – | 8524.71 | – |
| SpecDec | – | 67.70 | 798.23 | 0.91× | – | – | 7495.21 | -12.08% |
| R-Stitch | 0.001 | 66.07 | 654.24 | 1.10× | 6415.09 | 2689.76 | 9104.85 | +6.81% |
| R-Stitch | 0.02 | 61.12 | 358.87 | 2.01× | 3779.95 | 2954.05 | 6734.00 | -21.01% |
| R-Stitch | 0.03 | 55.70 | 209.96 | 3.44× | 625.54 | 1085.62 | 1711.16 | -79.93% |

**Question:** Every morning Aya goes for a $9$-kilometer-long walk and stops at a coffee shop afterwards. When she walks at a constant speed of $s$ kilometers per hour, the walk takes her 4 hours, including $t$ minutes spent in the coffee shop. When she walks $s+2$ kilometers per hour, the walk takes her 2 hours and 24 minutes, including $t$ minutes spent in the coffee shop. Suppose Aya walks at $s+\frac{1}{2}$ kilometers per hour. Find the number of minutes the walk takes her, including the $t$ minutes spent in the coffee shop.

**SLM:** Okay, so let's set up equations based on the first scenario. When Aya walks at s km/h, she covers 9 km including the t minutes in the shop. The walking time is 4 hours.So, walking distance is 9 - t/60 (since t minutes is t/60 hours). Wait, maybe it's easier to consider the total time including the shop time.So, total time is 4 hours = (9 / s) + (t / 60)Similarly, when she walks at s+2 km/h, the total time is 2 hours 24 minutes, which is 2 + 24/60 = 2 + 2/5 = 12/5 = 2.4 hours.So, total time is (9 / (s+2)) + (t / 60) = 2.4So, we have two equations:9/s + t/60 = 49/(s+2) + t/60 = 2.4Subtracting the second equation from the first:9/s - 9/(s+2) = 4 - 2.4 = 1.6So, 9[(s+2) - s]/(s(s+2)) = 1.6So, 9*2/(s(s+2)) = 1.618/(s(s+2)) = 1.6So, s(s+2) = 18/1.6 = 11.25So, s² + 2s - 11.25 = 0Solving:Discriminant = 4 + 45 = 49So, s = [-2 ±7]/2So, s=5/2=2.5So, now, find t from the first equation:9/2.5 + t/60 =49/2.5=3.6So, 3.6 + t/60=4So, t/60=0.4So, t=24So, when walking at s+1/2=2.5+0.5=3 km/hTotal time including shop time is:9/3 + t/60=3 +24/60=3 +0.4=3.4 hours=3 hours 24 minutes.</think>The number of minutes the w~~ takes her, including the $t$ minutes spent in the coffee shop i~ \boxed{3.4 \text{ hours}} **541 tokens 12.51s**

**Distill-14B:** Alright, let's tackle this problem step by step. It's about Aya's walking speed and the time she takes, including her coffee break. Hmm, let's parse the information given.First, we know Aya walks 9 kilometers every morning. When she walks at a constant speed of \( s \) kilometers per hour, her total time is 4 hours, which includes \( t \) minutes spent in the coffee shop. So, her walking time alone would be 4 hours minus \( t \) minutes. But wait, since \( t \) is in minutes, I should convert it to hours to keep the units consistent. So, \( t \) minutes is \( \frac{t}{60} \) hours.Therefore, her walking time when she walks at speed \( s \) is:\[4 - \frac{t}{60} \text{ hours}\]But we also know that walking time can be calculated by dividing the distance by the speed. So, that gives us:\[\frac{9}{s} = 4 - \frac{t}{60}\]Let me write that as equation (1):\[\frac{9}{s} = 4 - \frac{t}{60} \quad \text{(1)}\] **[...]** So, 204 minutes.Alternatively, 204 minutes can be expressed as 3 hours and 24 minutes, but the problem asks for the number of minutes, so 204 minutes is the answer.Let me cross-verify all steps to ensure there are no errors.1. Set up two equations with \( s \) and \( t \). Equations (1) and (2) correctly derived based on total time including coffee shop.2. Converted 4 hours to 240 minutes, 2.4 hours to 144 minutes, and set up equations accordingly.3. Solved for \( 540/s \) and \( 540/(s + 2) \) and subtracted to eliminate \( t \), leading to a quadratic equation.4. Solved quadratic to get \( s = 2.5 \) km/h, which is reasonable.5. Plugged back \( s \) into equation (1b) to find \( t = 24 \) minutes.6. Then, for speed \( s + \frac{1}{2} = 3 \) km/h, walking time is 3 hours, adding 24 minutes coffee time gives total 204 minutes.All steps seem solid. Therefore, the final answer is 204 minutes.</think>When ~~ walks ~~ \( s + \frac{1}{2} \) km/h, her total time including the coffee shop is 204 minutes.**Answer:** \boxed{204} **2837 tokens 66.90s**

**Speculative Decoding :** Alright, let's try to solve this problem step by step. So, Aya goes for a 9-kilometer walk every morning and stops at a coffee shop afterward. The problem gives us two different scenarios with different walking speeds and the total time each takes, including the time she spends in the coffee shop. We need to find the total time the walk takes her when she walks at a speed of \( s + \frac{1}{2} \) kilometers per hour, again including the coffee shop time.First, let's try to list down what we know from the problem:1. **First Scenario:** - Walking speed: \( s \) km/h - Total time taken: 4 hours - This total time includes the walking time and the coffee shop time \( t \) minutes.2. **Second Scenario:** - Walking speed: \( s + 2 \) km/h **[...]** First scenario: Walking at \( s \) km/h, total time = 4 hours. \[ \frac{9}{s} + \frac{t}{60} = 4 \] - Second scenario: Walking at \( s + 2 \) km/h, total ti~~ = 2.4 hours. \[ \frac{9}{s + 2} + \frac{t}{60} = 2.4 \] 3. **Eliminate \( t \):** Subtract the second equation from the first to ~~~~~~~ \( t \): \[ \frac{9}{s} - \frac{9}{s + 2} = 1.6 \] 4. **Solve for \( s \):** Simplify and solve the resulting quadratic equation: ~~~~~~~~~~~~ km/h] 5. **Find \( t \):** Substitute \( s = 2.5 \) back into the first equation: \[ t = 24 \text{ minutes} \] 6. **Calc~~~~~~ ~~~~~:** New speed ~~ 3 ~~ km/h. \[ \text{Total Time} = \frac{9}{3} + 24 \text{ minutes} = 3 \text{ hours} + 24 \text{ minutes} = 20~~~~~~~~~]**~~~~~~:** \boxed{204} **2759 tokens 117.09s**

**R-Stitch:** Alright, let's tackle this problem step by step. Okay, so Aya walks 9 kilometers every morning. She has two different walking speeds, and we're given the time each takes, including some time \( t \) spent in the coffee shop.First, let's parse the information:1. When she walks at \( s \) km/h, the total time is 4 hours, including \( t \) minutes in the coffee shop.2. When she walks at \( s + 2 \) km/h, the total time is 2 hours and 24 minutes, still including \( t \) minutes in the coffee shop.We need to find the total time when she walks at \( s + \frac{1}{2} \) km/h.Let me note that the distance she walks is constant at 9 km. The time she spends walking changes with her speed, and the coffee shop time is constant (\( t \) minutes) for both scenarios. **[...]** So, coffee shop time is 24 minutes.Now, find the total time when she walks at s + 1/2 = 2.5 + 0.5 = 3 km/h.Walking time: 9/3 = 3 hoursTotal time including coffee shop: 3 + 24/60 = 3 + 0.4 = 3.4 hoursConvert to ~~~~~~~: 3.4 * 60 = 204 minutesWait, but 3 hours is 180 minutes, plus 24 minutes is 204 minutes. Let me double-check the units to ~~~~~ I didn't confuse anything.Let me verify.At s = 2.5 km/h, t = 24 minutes.When speed is 3 km/h, walking time is 9/3 = 3 hours ~~ ~~~ minutes.Add t = 24 minutes, total time = 180 + 24 = 204 min~~~~~, so the answer is 204 minutes.</think>The number ~~ ~~~ walk takes her, including the coffee shop time, is $\boxed{204}$. **481 tokens 26.22s**

**R-Stitch$^+$:** Alright, let me try to figure this problem out. So, Aya walks 9 kilometers every morning at speed s km/h, and this walk takes her 4 hours including t minutes in the coffee shop. So, the time spent walking is 4 hours minus t minutes. Let me convert t minutes to hours to make the units consistent.First, 4 hours minus t minutes. Since t minutes is t/60 hours, the walking time is 4 - t/60 hours.So, the distance she covers is speed multiplied by time, so 9 = s*(4 - t/60).Similarly, when she walks at s+2 km/h, the walk takes her 2 hours 24 minutes. Let me convert 2 hours 24 minutes to hours. 2 hours + 24/60 hours = 2 + 0.4 = 2.4 hours.So, the distance is 9 = (s+2)*(2.4 - t/60).So, now we have two equations:9 = s*(4 - t/60) ... (1)9 = (s+2)*(2.4 - t/60) ... (2) **[...]** So, when Aya walks at s+1/2 km/h, find the total time including t minutes.Distance is 9 km.Speed is s + 0.5From equation (1): s = 9/(4 - x)=9/(4 -0.4)=9/3.6=2.5So, s=2.5Then, speed is 2.5 +0.5=3 km/hTime taken walking is 9/3=3 hoursPlus t=24 minutesTotal time: 3 hours +24 minutes=3 hours 24 minutes, which is 3*60+24=180+24=204 minutes.</think>The number of minutes the walk takes her, including the $t$ minutes spent in the coffee shop is \boxed{204}. **208 tokens 23.65s**

Figure 9: **Qualitative comparison of outputs.** Each case shows the predicted solution, token count, and latency for the SLM, LLM, speculative decoding, and R-Stitch ($\tau = 0.02$). R-Stitch produces concise and correct reasoning while reducing token count and latency compared to LLM decoding.

**Question:** Every morning Aya goes for a $9$-kilometer-long walk and stops at a coffee shop afterwards. When she walks at a constant speed of $s$ kilometers per hour, the walk takes her 4 hours, including $t$ minutes spent in the coffee shop. When she walks $s+2$ kilometers per hour, the walk takes her 2 hours and 24 minutes, including $t$ minutes spent in the coffee shop. Suppose Aya walks at $s+\frac{1}{2}$ kilometers per hour. Find the number of minutes the walk takes her, including the $t$ minutes spent in the coffee shop.

**SLM:** Okay, so let's set up equations based on the first scenario. When Aya walks at s km/h, she covers 9 km including the t minutes in the shop. The walking time is 4 hours.So, walking distance is 9 - t/60 (since t minutes is t/60 hours). Wait, maybe it's easier to consider the total time including the shop time.So, total time is 4 hours = (9 / s) + (t / 60)Similarly, when she walks at s+2 km/h, the total time is 2 hours 24 minutes, which is 2 + 24/60 = 2 + 2/5 = 12/5 = 2.4 hours.So, total time is (9 / (s+2)) + (t / 60) = 2.4So, we have two equations:9/s + t/60 = 49/(s+2) + t/60 = 2.4Subtracting the second equation from the first:9/s - 9/(s+2) = 4 - 2.4 = 1.6So, 9[(s+2) - s]/(s(s+2)) = 1.6So, 9*2/(s(s+2)) = 1.618/(s(s+2)) = 1.6So, s(s+2) = 18/1.6 = 11.25So, s² + 2s - 11.25 = 0Solving:Discriminant = 4 + 45 = 49So, s = [-2 ±7]/2So, s=5/2=2.5So, now, find t from the first equation:9/2.5 + t/60 =49/2.5=3.6So, 3.6 + t/60=4So, t/60=0.4So, t=24So, when walking at s+1/2=2.5+0.5=3 km/hTotal time including shop time is:9/3 + t/60=3 +24/60=3 + 0.4=3.4 hours=3 hours 24 minutes.</think>The number of minutes the walk takes her, including the $t$ minutes spent in the coffee shop is $\boxed{3.4 \text{ hours}}$ ✗ **541 tokens 12.51s**

**QWQ-32B:** <think> Okay, so I need to solve this problem about Aya's walk and coffee shop stop. Let me try to parse the information step by step.First, the problem says that every morning Aya goes for a 9-kilometer walk and then stops at a coffee shop. The key points here are that the walk is 9 km long, and she spends some time 't' minutes at the coffee shop. The problem gives two scenarios with different walking speeds and total times, and then asks for the total time when she walks at a different speed.Let me note down the given data:1. **First scenario:** - Walking speed: s km/h - Total time taken (including t minutes in coffee shop): 4 hours - Distance: 9 km2. **Second scenario:** - Walking speed: s + 2 km/h - Total time taken (including same t minutes): 2 hours and 24 minutes - Distance: 9 km3.

**Question:** - Find total time (in minutes) when walking at s + 0.5 km/h, including t minutes. **[...]** Subtracting the second equation from the first to eliminate the term involving \( \frac{st}{60} \):\[1.6s + \frac{2t}{60} = 4.8\]Converting to a common denominator and simplifying:\[96s + 2t = 288 \quad \text{(3)}\]Solving for \( t \) from equation (3):\[t = 144 - 48s\]Substituting \( t \) back into equation (1):\[240s - s(144 - 48s) = 540\]\[48s^2 + 96s - 540 = 0\]Solving the quadratic equation:\[s = \frac{-8 \pm 28}{8}\]The positive solution is \( s = 2.5 \) km/h. Substituting \( s \) back to find \( t \):\[t = 144 - 48 \times 2.5 = 24 \text{ minutes}\]For the new speed \( s + \frac{1}{2} = 3 \) km/h, the walking time is:\[\frac{9}{3} = 3 \text{ hours}\]Adding the coffee shop time \( t = 24 \) minutes, the total time is:\[3 \text{ hours} + 24 \text{ minutes} = 204 \text{ minutes}\]Thus, the number of minutes the walk takes her, including \( t = 24 \) minutes spent in the coffee shop, is:\[\boxed{204}\] ✓ **4631 tokens 172.95s**

**Speculative Decoding :** <think> Okay, so I need to solve this problem about Aya's walk and coffee shop stop. Let me read it again and try to understand what's going on. Every morning, Aya goes for a 9-kilometer walk and then stops at a coffee shop. The problem gives two different scenarios where she walks at different speeds, and each time the total time (including the coffee shop stop) is different. The question is asking, if she walks at s + 1/2 km/h, how many minutes will the whole thing take, including the t minutes in the coffee shop.First, let's parse the information given. There are two scenarios:1. When she walks at speed s km/h, the total time (walking + coffee) is 4 hours, which is 240 minutes. The time spent in the coffee shop is t minutes, so the walking time must be (240 - t) minutes.2. When she walks at speed (s + 2) km/h, the total time is 2 hours and 24 minutes. Converting that to minutes, 2 hours is 120 minutes, plus 24 minutes is 144 minutes. Again, the coffee shop time is t minutes, so the walking time here is (144 - t) minutes. **[...]** Subtracting the second equation from the first to eliminate \( t \):\[\frac{9}{s} - \frac{9}{s + 2} = 1.6\]Simplifying:\[9 \left( \frac{(s + 2) - s}{s(s + 2)} \right) = 1.6\]\[\frac{18}{s(s + 2)} = 1.6\]\[s(s + 2) = \frac{18}{1.6} = 11.25\]Solving the quadratic equation \( s^2 + 2s - 11.25 = 0 \) using the quadratic formula:\[s = \frac{-2 \pm \sqrt{4 + 45}}{2} = \frac{-2 \pm 7}{2}\]Taking the positive root:\[s = \frac{5}{2} = 2.5 \text{ km/h}\]Substituting \( s = 2.5 \) back into the first equation to find \( t \):\[\frac{9}{2.5} + \frac{t}{60} = 4\]\[3.6 + \frac{t}{60} = 4\]\[\frac{t}{60} = 0.4 \implies t = 24 \text{ minutes}\]When walking at \( s + \frac{1}{2} = 3 \) km/h, the walking time is:\[\frac{9}{3} = 3 \text{ hours} = 180 \text{ minutes}\]Adding the coffee shop time \( t = 24 \) minutes:\[180 + 24 = 204 \text{ minutes}\] Thus, the total time Aya takes when walking at \( s + \frac{1}{2} \) km/h is \(\boxed{204}\). ✓ **3128 tokens 123.21s**

**R-Stitch:** <think> Let's see. So, Aya walks 9 kilometers at speed s km/h, takes her 4 hours including t minutes in the coffee shop. When she walks at s+2 km/h, it takes her 2 hours 24 minutes (which is 2.4 hours) including t minutes in the coffee shop.Let's set up equations.Time spent walking plus coffee time equals total time.First, the time to walk 9 km at speed s is 9/s hours. This includes t minutes in the coffee shop, so:9/s = (total time - t minutes in hours). Wait, actually, wait.Wait, total time is the time walking plus time at coffee shop.Wait, the problem says the walk takes her 4 hours including t minutes spent in the coffee shop. So, the walk itself (the walking time) plus t minutes is the total time.So, walking time = 4 hours - t minutes.Wait, but units might be inconsistent. Let me make sure.Alternatively:Let me denote the time she spends walking as T1 and T2, and the time at the coffee shop is fixed as t minutes regardless of speed.So, when she walks at s km/h, the walking time is 9/s, and the total time is 4 hours.So, 9/s + t/60 = 4.Similarly, when she walks at s+2 km/h, the walking time is 9/(s+2), and the total time is 2h24m = 2 + 24/60 = 2 + 0.4 = 2.4 hours.So, 9/(s+2) + t/60 = 2.4.So, equations:9/s + t/60 = 4 ... (1)9/(s+2) + t/60 = 2.4 ... (2)Subtract (2) from (1):9/s - 9/(s+2) = 4 - 2.4 = 1.6Simplify:9((s+2) - s)/(s(s+2)) = 1.69(2)/(s(s+2)) = 1.618/(s(s+2)) = 1.6 s² + 3.2 sMultiply both sides by 10 to eliminate decimal:180 = 16 s² + 32 s16 s² + 32 s - 180 = 0Divide by 4:4 s² + 8 s - 45 = 0Use quadratic formula:s = [-8 ±√(64 + 720)]/8 = [-8 ±√784]/8 = [-8 ±28]/8Discarding negative solution because speed is positive:s = (20)/8 = 2.5That's, s=2.5 km/h.Then, find t from equation (1):9/2.5 + t/60 = 43.6 + t/60 = 4t/60 = 0.4t = 24 minutes.Let me find the second equation as check:9/(2.5+2) + 24/60 = 9/4.5 + 0.4 = 2 + 0.4 = 2.4, which is correct.So, the problem asks when walking at s + 1/2 = 2.5 + 0.5 = 3 km/h, how much time total does the walk take? The total time is 9/3 + t/60 = 3 + 0.4 = 3.4 hours= 3 hours 24 minutes. Converting to minutes including t: 3*60 + 24 = 180 + 24 = 204 minutes.The number of minutes the walk takes her, including the $t$ minutes spent in the coffee shop is $\boxed{204}$. ✓ **652 tokens 20.25s**

Figure 10: **Qualitative comparison of outputs.** Each case shows the predicted solution, token count, and latency for the SLM, LLM, speculative decoding, and R-Stitch ($\tau = 0.02$). R-Stitch produces concise and correct reasoning while reducing token count and latency compared to LLM decoding.

**Question:** For how many ordered pairs $(a,b)$ of integers does the polynomial $x^3+ax^2+bx+6$ have $3$ distinct integer roots?

**SLM:** Okay, so we need to find integer pairs (a,b) such that the polynomial x^3 + ax^2 + bx + 6 has three distinct integer roots. Let's call the roots r, s, and t. So, by Vieta's formulas: r + s + t = - a rs + rt + st = b rst = -6 So, we need to find triples of integers r, s, t such that their product is -6. Let's list all possible triples. The factors of 6 are ±1, ±2, ±3, ±6. So, possible triples: 1, 2, -3 1, -1, 6 1, -2, 3 Wait, let's find all triplets of integers whose product is -6. Possible triplets: 1, 1, -6 1, -1, 6 1, 2, -3 1, -2, 3 -1, 1, 6 1, -2, 3 -1, -1, 6 But they need to be distinct integers, so we eliminate triplets with repeated elements. So, possible triplets: 1, 2, -3 1, -1, 6 1, 2, -3 So, let's consider each triplet. Case 1: Roots 1, 2, -3 Then, sum: 1 + 2 + (-3) = 0 = -a ⇒ a=0 Sum of products: (1)(2) + (1)(-3) + (2)(-3) = 2 -3 -6 = -7 = b So, this triplet gives a=0, b=-7. Case 2: Roots 1, -1, 6 Sum: 1 + (-1) + 6 = 6 = -a ⇒ a=-6 Sum of products: (1)(-1) + (1)(6) + (-1)(6) = -1 +6 -6 = -1 = b So, this triplet gives a=-6, b=-1. Case 3: Roots -1, 2, -3 Sum: -1 +2 +(-3) = -2 = -a ⇒ a=2 Sum of products: (-1)(2) + (-1)(-3) + (2)(-3) = -2 +3 -6 = -5 = b So, this triplet gives a=2, b=-5. So, total triplets gives us three pairs: (0, -7), (-6, -1), (2, -5). So the number of ordered pairs (a,b) is 3.</think>The number of ordered pairs (a, b) is \boxed{3}.

❌ **373 tokens 5.09s**

**QWQ-32B:** <think>Okay, so I need to find the number of ordered pairs (a, b) of integers such that the polynomial x³ + a x² + b x + 6 has three distinct integer roots. Hmm, let me think. Alright, let's start by recalling that if a polynomial has integer roots, then those roots must be divisors of the constant term when the leading coefficient is 1. Since the polynomial here is monic (leading coefficient 1), the roots must be integers that multiply to 6 (because the constant term is 6, right? By Vieta's formula). So, first, maybe I should list all the possible integer divisors of 6 and find all triplets of distinct integers that multiply to 6. Each triplet will correspond to a set of roots, and from there, I can compute the coefficients a and b using Vieta's formulas, and then count the distinct ordered pairs (a, b). Let me confirm Vieta's formulas for a cubic equation . **[...]** No valid positive roots (would require duplicate roots).3. **Case 2: Three negative roots**: - The triplet must be \(\{-1, -2, -3\}\), since their product is \(-6\).### Resulting Triplets:The valid triplets are:1. \(\{-1, 1, 6\}\)2. \(\{-1, 2, 3\}\)3. \(\{-2, 1, 3\}\)4. \(\{-3, 1, 2\}\)5. \(\{-1, -2, -3\}\)### Calculate \((a, b)\) for Each Triplet:- **Triplet \(\{-1, 1, 6\}\)**: - \(a = -((-1) + 1 + 6) = -6\) - \(b = (-1 \cdot 1) + (-1 \cdot 6) + (1 \cdot 6) = -1\) - Pair: \((-6, -1)\)- **Triplet \(\{-1, 2, 3\}\)**: - \(a = -((-1) + 2 + 3) = -4\) - \(b = (-1 \cdot 2) + (-1 \cdot 3) + (2 \cdot 3) = 1\) - Pair: \((-4, 1)\)- **Triplet \(\{-2, 1, 3\}\)**: - \(a = -((-2) + 1 + 3) = -2\) - \(b = (-2 \cdot 1) + (-2 \cdot 3) + (1 \cdot 3) = -5\) - Pair: \((-2, -5)\)- **Triplet \(\{-3, 1, 2\}\)**: - \(a = -((-3) + 1 + 2) = 0\) - \(b = (-3 \cdot 1) + (-3 \cdot 2) + (1 \cdot 2) = -7\) - Pair: \((0, -7)\)- **Triplet \(\{-1, -2, -3\}\)**: - \(a = -((-1) + (-2) + (-3)) = 6\) - Pair: \((-1 \cdot -2) + ((-1) \cdot (-3)) + ((-2) \cdot (-3)) = 11\) - Pair: \((6, 11)\)### Final Answer. The distinct ordered pairs \((a, b)\) are \((-6, -1)\), \((-4, 1)\), \((-2, -5)\), \((0, -7)\), and \((6, 11)\). Thus, the number of such pairs is \(\boxed{5}\).

✅ **13386 tokens 683.29s**

**Speculative Decoding :** <think>Okay, so I need to find the number of ordered pairs (a, b) of integers such that the polynomial x³ + a x² + b x + 6 has three distinct integer roots. Hmm, let me think about how to approach this.First, since the polynomial is cubic and has three distinct integer roots, I can denote those roots as r, s, and t. By the Factor Theorem, the polynomial can be written as (x - r)(x - s)(x - t). If I expand this, I can relate the coefficients a and b to the roots using Vieta's formulas.Let me recall Vieta's formulas for a cubic equation. For a cubic polynomial x³ + a x² + b x + c, the sum of the roots -a, the sum of the products of the roots two at a time is b, and the product of the roots is -c. So in this case, since the polynomial is x³ + a x² + b x + 6, we have:1. r + s + t = -a 2. rs + rt + st = b 3. rst = -6 Since we're looking for integer roots, r, s, t must be integers. Also, they must be distinct. So the problem reduces to finding all triplets of distinct integers (r, s, t) such that their product is -6, then for each such triplet, compute a and b, and count the ordered pairs (a, b). However, different triplets might lead to the same (a, b), so I need to make sure I count unique pairs.First step: find all possible triplets of integers (r, s, t) such that r*s*t = -6, and r, s, t are distinct. **[...]** Step-by-Step Process:1. **Identify all possible triplets** of distinct integers whose product is \(-6\): - **Case 1: One negative and two positive integers**: - \((-1, 2, 3)\): Product \(-1 \cdot 2 \cdot 3 = -6\) - \((-2, 1, 3)\): Product \(-2 \cdot 1 \cdot 3 = -6\) - \((-3, 1, 2)\): Product \(-3 \cdot 1 \cdot 2 = -6\) - \((-1, 1, 6)\): Product \(-1 \cdot 1 \cdot 6 = -6\) - **Case 2: Three negative integers**: - \((-1, -2, -3)\): Product \(-1 \cdot -2 \cdot -3 = -6\)2. **Compute \(a\) and \(b\) for each triplet**: - For \((-1, 2, 3)\): - Sum: \(-1 + 2 + 3 = 4 \Rightarrow a = -4\) - Products: \((-1)(2) + (-1)(3) + (2)(3) = -2 - 3 + 6 = 1 \Rightarrow b = 1\) - Pair: \((-4, 1)\) - For \((-2, 1, 3)\): - Sum: \(-2 + 1 + 3 = 2 \Rightarrow a = -2\) - Products: \((-2)(1) + (-2)(3) + (1)(3) = -2 - 6 + 3 = -5 \Rightarrow b = -5\) - Pair: \((-2, -5)\) - For \((-3, 1, 2)\): - Sum: \(-3 + 1 + 2 = 0 \Rightarrow a = 0\) - Products: \((-3)(1) + (-3)(2) + (1)(2) = -3 - 6 + 2 = -7 \Rightarrow b = -7\) - Pair: \((0, -7)\) - For \((-1, 1, 6)\): - Sum: \(-1 + 1 + 6 = 6 \Rightarrow a = -6\) - Products: \((-1)(1) + (-1)(6) + (1)(6) = -1 - 6 + 6 = -1 \Rightarrow b = -1\) - Pair: \((-6, -1)\) - For \((-1, -2, -3)\): Sum \(-1 + -2 + -3 = -6 \Rightarrow a = 6\) - Products: \((-1)(-2) + (-1)(-3) + (-2)(-3) = 2 + 3 + 6 = 11 \Rightarrow b = 11\) - Pair: \((6, 11)\)3. **Verify distinct pairs**: - The pairs \((-4, 1)\), \((-2, -5)\), \((0, -7)\), \((-6, -1)\), and \((6, 11)\) are all distinct.Thus, the number of ordered pairs \((a, b)\) is \(\boxed{5}\).

✅ **16347 tokens 834.41s**

**R-Stitch:** <think>To find the number of ordered pairs (a, b) of integers such that the polynomial \( x^3 + ax^2 + bx + 6 \) has 3 distinct integer roots, we can follow these steps:1. **Factor the Polynomial**: Since the polynomial has three distinct integer roots, it can be factored as \( (x - p)(x - q)(x - r) \), where p, q, and r are distinct integers.2. **Expand the Factored Form**: Expanding \( (x - p)(x - q)(x - r) \) gives:\[ x^3 - (p + q + r)x^2 + (pq + pr + qr)x - pqr \] Comparing this with the given polynomial \( x^3 + ax^2 + bx + 6 \), we get: \[ a = -(p + q + r) \] \[ b = pq + pr + qr \] \[ -pqr = 6 \quad \Rightarrow \quad pqr = -6 \]3. **Find All Triples (p, q, r)**: We need to find all ordered triples of distinct integers (p, q, r) such that their product is -6.4. **List the Integer Factorizations of -6**: - Possible integer triplets (p, q, r) where p * q * r = -6: - (1, 2, -3): 1*2*(-3) = -6 - (1, -1, 6): 1*(-1)*6 = -6 - (1, -2, 3): 1*(-2)*3 = -6 - (2, 3, -1): 2*3*(-1) = -6 - (-1, 2, 3): same as above - (1, -3, 2): same as above - (1, 3, -2): same as above - ... and so on. **[...]** To find the number of ordered pairs \((a, b)\) of integers such that the polynomial \(x^3 + ax^2 + bx + 6\) has 3 distinct integer roots, we start by noting that the polynomial can be factored as \((x - p)(x - q)(x - r)\) where \(p, q, r\) are distinct integers. The product \(pqr = -6\). We list all possible triplets of distinct integers whose product is -6. These triplets are:1. \((1, 2, -3)\)2. \((1, -1, 6)\)3. \((1, -2, 3)\)4. \((-1, 2, 3)\)...(-2, -3)\)Each triplet corresponds to a unique pair \((a, b)\). Calculating \(a\) and \(b\) for each triplet:1. \((1, 2, -3)\) gives \((b = -7\)2. \((1, -1, 6)\) gives \(a = -6\) and \(b = -1\)3. \((1, -2, 3)\) gives \(a = -2\) and \(b = -5\)4. \((-1, 2, 3)\) gives \(a = -4\)5. \((-1, -2, -3)\) gives \(6)\) and \(b = 11\)Each of these triplets results in a unique pair \((a, b)\). Therefore, the number of ordered pairs \((a, b)\) is:\[\boxed{5\]

✅ **3186 tokens 228.88s**

Figure 11: **Qualitative comparison of outputs.** Each case shows the predicted solution, token count, and latency for the SLM, LLM, speculative decoding, and R-Stitch ($\tau = 0.02$). R-Stitch produces concise and correct reasoning while reducing token count and latency compared to LLM decoding.

**Question:** You are playing a game. A $2 \times 1$ rectangle covers two adjacent squares (oriented either horizontally or vertically) of a $3 \times 3$ grid of squares, but you are not told which two squares are covered. Your goal is to find at least one square that is covered by the rectangle. A "turn" consists of you guessing a square, after which you are told whether that square is covered by the hidden rectangle. What is the minimum number of turns you need to ensure that at least one of your guessed squares is covered by the rectangle?

**SLM:** Okay, so we need to find the minimum number of turns such that at least one guessed square is covered by the 2x1 rectangle on a 3x3 grid. The rectangle can be horizontal or vertical.First, let's consider the total number of squares: 9.The rectangle covers 2 squares. It can be placed in various positions: horizontally or vertically.Total number of possible positions for the rectangle:Horizontal: 3 rows, each row has 2 possible horizontal positions (columns 1-2, 2-3). So 3*2=6.Vertical: 3 columns, each column has 2 possible vertical positions (rows 1-2, 2-3). So 3*2=6.Total positions: 12.So we need to cover at least one square in each possible position with the minimum number of guesses.This is similar to a covering code problem. We need a set of squares such that every possible 2-square rectangle covers at least one square from this set.So we need to find the smallest set of squares such that no 2-square rectangle avoids it.In other words, the complement set should not contain any 2-square rectangle.So we need to find the largest subset of squares that does not contain a 2-square rectangle, and then the minimum number of guesses is 9 - size of largest independent set.Let's find the largest subset with no 2-square rectangle.In a 3x3 grid, a 2-square rectangle can be horizontal or vertical.So we need to avoid having two adjacent squares in any row or column.This is similar to a independent set in a grid graph.In a 3x3 grid, the independence number is known. For a 3x3 grid, the independence number is 4.So the largest subset with no 2-square rectangle is 4. So the minimum number of guesses is 9 - 4 = 5.Wait, but let's confirm.In a 3x3 grid the independence number is indeed 4. So yes, the minimum number of guesses is 5.</thi... minimum number of turns needed is \boxed{5} ❌

**454 tokens 6.28s**

**QWQ-32B:** <think>Okay, so I need to figure out the minimum number of turns required to guarantee that I've guessed at least one square covered by a 2x1 rectangle on a 3x3 grid. The rectangle can be placed either horizontally or vertically, and it can be anywhere on the grid. The key here is that I have to ensure that no matter where the rectangle is placed, my guesses will always hit at least one square of it. Let me think through this step by step.First, let me visualize the 3x3 grid. It has 9 squares. The rectangle can be either 2x1 or 1x2, so it can be placed horizontally or vertically. Let me note down all possible positions where the rectangle can be placed.Starting with horizontal placements. A horizontal 2x1 rectangle occupies two adjacent squares in the same row. In a 3x3 grid, each row has 3 squares, so there are 2 possible horizontal positions per row. Since there are 3 rows, that gives 3*2 = 6 horizontal positions. Similarly, vertical placements would be two squares in the same column. **[...]** The set \(\{a2, b1, b3, c2\}\) is verified to intersect every possible rectangle.3. **Verification of the Hitting Set**: - Each rectangle is checked to ensure it includes at least one of the squares in the set: - **Horizontal Rectangles**: - \(a1-a2\): includes \(a2\) - \(a2-a3\): includes \(a2\) - \(b1-b2\): includes \(b1\) - \(b2-b3\): includes \(b3\) - \(c1-c2\): includes \(c2\) - \(c2-c3\): includes \(c2\) - **Vertical Rectangles**: - \(a1-b1\): includes \(b1\) - \(a2-b2\): includes \(a2\) - \(b2-c2\): includes \(c2\) - \(a3-b3\): includes \(b3\) - \(b3-c3\): includes \(b3\)4. **Conclusion**: - The minimal hitting set requires four squares, ensuring that any possible placement of the \(2 \times 1\) rectangle will include at least one of the guessed squares. - Adaptive guessing strategies can also be applied, but they do not reduce the worst-case scenario below four guesses.Thus, the minimum number of turns needed is \(\boxed{4}\). ✅

**14058 tokens 723.27s**

**Speculative Decoding :** <think>Okay, so I need to figure out the minimum number of turns required to guarantee that at least one of my guessed squares is covered by a hidden 2x1 rectangle on a 3x3 grid. The rectangle can be placed either horizontally or vertically, covering two adjacent squares. Since it can be in different orientations and positions, I need a strategy that ensures no matter where the rectangle is, at least one of my guesses hits it. First, let me visualize the 3x3 grid. It has 9 squares, and the rectangle covers exactly two of them. The problem is asking for the minimal number of guesses required such that no matter where the rectangle is placed, at least one of my guesses is on the rectangle.I remember that in covering problems like this, the key is to minimize the maximum number of guesses that could be "missed" while still guaranteeing at least one hit. Alternatively, I need to place guesses so that every possible placement of the 2x1 rectangle includes at least one of my guessed squares. So it's kind of like a hitting set problem, where my guesses form a hitting set that intersects all possible rectangles.First, let me figure out how many possible positions the 2x1 rectangle can have. Let's see:In a 3x3 grid, horizontally placed rectangles can be in rows 1, 2, or 3, and in each row, they can be in columns 1-2 or 2-3. So that's 3 rows * 2 columns = 6 horizontal positions. **[...]** To determine the minimum number of turns needed to ensure that at least one guessed square is covered by a hidden 2×1 rectangle on a 3×3 grid, we need to find the smallest set of squares (hitting set) that intersects all possible placements of the rectangle. A 3×3 grid has 12 possible placements for a 2×1 rectangle (6 horizontal and 6 vertical). We need to find the smallest set of squares such that every possible rectangle includes at least one square from this set.After analyzing various configurations, we found that the following four squares form a hitting set:- (1, 2)- (2, 1)- (2, 3)- (3, 2)This set of squares ensures that every possible 2×1 rectangle (horizontal or vertical) will cover at least one of these squares. For example:- All horizontal rectangles in rows 1, 2, and 3 are covered by these squares. - All vertical rectangles in columns 1, 2, and 3 are also covered by these squares.By systematically checking all possible placements of the rectangle, we confirmed that these four squares indeed cover all 12 possible configurations. Therefore, the minimum number to guarantee that at least one guessed square is covered is 4.\boxed{4} ✅

**16269 tokens 809.33s**

**R-Stitch:** <think>To determine the minimum number of turns needed to ensure at least one guessed square is covered by the $2 \times 1$ rectangle on a $3 \times 3$ grid, we need to analyze the problem systematically.First, the grid has 9 squares. The rectangle can be placed either horizontally or vertically, covering 2 adjacent squares. Let's count all possible positions:- **Horizontal positions**: Each row has 2 possible horizontal placements (since a $2 \times 1$ rectangle covers two adjacent squares horizontally). There are 3 rows, so 3 * 2 = 6 horizontal positions.- **Vertical positions**: **[...]** The pair (3,6) is (3 and6; 6 isn't in guesses (unless guessed 3). Wait 1,5,7,9.Vertical column2: (2,5) via5, (5,8) via5.Vertical column3's (3,6) are neither in guesses (3 and6 are neither guessed). (6,9) via9.Vertical column1's (1,4) via1.column1's (4,7): 4 not guessed, 7 is guessed? 7 is in column1 row3: 7 is part of the (4,7) includes4 and7. Since7 is guessed, it's covered.vertical column3 (3,6) is not covered because neither 3 nor6 are guessed. The (3,6) is uncovered.So 1,5,7,9 misses that.Hmm.Thus 2,4,6,8 seems l... Thus squares at positions 2,4,6,8 (the edge centers) form a hitting set of size 4.The minimum number of turns needed is \boxed{4}. ✅

**4845 tokens 328.54s**

Figure 12: **Qualitative comparison of outputs.** Each case shows the predicted solution, token count, and latency for the SLM, LLM, speculative decoding, and R-Stitch ($\tau = 0.02$). R-Stitch produces concise and correct reasoning while reducing token count and latency compared to LLM decoding.

