# OpenReview forum: "R-Stitch: Dynamic Trajectory Stitching for Efficient Reasoning"
_ICLR.cc/2026/Conference — Submitted to ICLR 2026_

### Official Review · Reviewer_saCs · 2025-10-27

**Soundness:** 2
**Presentation:** 3
**Contribution:** 2
**Rating:** 2
**Confidence:** 4

**Summary:**

The paper proposes R-Stitch, a training-free hybrid decoding method that accelerates chain-of-thought reasoning by dynamically switching between a small language model and a large language model based on token entropy. Low-entropy tokens are handled by the SLM to save computation, while high-entropy tokens are delegated to the LLM to preserve accuracy. It shows that token entropy correlates with reasoning errors and uses it for token-level routing. In addition, the paper extends R-Stitch to R-Stitch+, which learns an adaptive switching policy via reinforcement learning with a latency-aware reward. Evaluation shows seepdup in generation with minimal accuracy loss on tested benchmarks.

**Strengths:**

- The paper tackles an important and timely problem: optimizing reasoning-time inference in large language models. As reasoning chains become longer and more expensive, improving efficiency without sacrificing accuracy is crucial for practical deployment.

- The use of entropy as a routing signal is well-motivated and empirically justified. Figure 3 effectively shows that incorrect outputs correlate with higher entropy, that most tokens have near-zero entropy, and that errors cluster around locally uncertain regions. This analysis builds a foundation for using entropy as a measure to guide dynamic switching.

- The approach leverages model-internal signals to optimize system-level behavior, is conceptually elegant.

**Weaknesses:**

- The novelty is limited. The method is conceptually incremental to speculative decoding, differing mainly in using entropy for routing rather than token-level agreement. Moreover, the connection between entropy, confidence, and uncertainty has been explored in prior work [1], yet this paper neither cites nor compares against them.

- The evaluation setting (batch size = 1) is unrealistic and potentially misleading. In real deployments, systems run with batch sizes > 1 to maximize GPU utilization, especially in frameworks like vLLM, which fully support batching. Running with batch = 1 may exaggerate latency gains, and the authors should justify this design choice.

- The evaluation scope is narrow. All experiments and entropy analyses are restricted to mathematical reasoning tasks. Since the paper claims general acceleration of chain-of-thought reasoning, results across other reasoning domains (e.g., code, logical reasoning, or QA) would make the findings more convincing.

- The paper ignores the memory overhead of maintaining both the LLM and SLM simultaneously. This dual-model setup increases GPU memory consumption and can reduce the number of requests processed in parallel. It’s possible that this constraint is why the experiments use batch size = 1, which should have been discussed explicitly.

- The method is not compared to alternative reasoning optimization approaches such as early termination or compression-based techniques, which also shorten reasoning traces. A fair evaluation should include or discuss these baselines to clarify the relative contribution.

- The approach’s robustness to entropy misprediction is unclear. If entropy fails to reflect uncertainty accurately (e.g., when the SLM is overconfident in incorrect outputs), the routing may degrade accuracy or fail to switch appropriately.

- The computational overhead of computing token entropy is not thoroughly analyzed. Although entropy can be obtained from logits, its per-token computation and switching logic could add nontrivial latency, which should be measured or discussed.

[1] Efficiently serving llm reasoning programs with certaindex. Fu, Yichao ; Chen, Junda ; Zhu, Siqi ; Fu, Zheyu ; Dai, Zhongdongming ; Zhuang, Yonghao ; Ma, Yian ; Qiao, Aurick ; Rosing, Tajana ; Stoica, Ion ; Zhang, Hao

**Questions:**

See the weaknesses section.

---

> ### Author Response · Authors · 2025-11-21
> **Response to Reviewer saCs**
>
> **Q1.** The novelty is limited. The method is conceptually incremental to speculative decoding, differing mainly in using entropy for routing rather than token-level agreement. Moreover, the connection between entropy, confidence, and uncertainty has been explored in prior work.
>
> **A1.** Below we clarify why our formulation is not just an incremental variant of speculative decoding.
> 1. **Speculative decoding relies on strict token-level consistency**, which prevents using concise and compact open-source SLMs. **R-Stitch removes this requirement** and routes high-uncertainty spans to the LLM while letting the SLM handle low-entropy regions, shifting the goal from enforcing consistency to **exploiting uncertainty**.
> 2. Prior entropy-based work[1] focuses on **early exiting within a single model**, assuming monotonic confidence growth. R-Stitch is different: it coordinates two models with distinct reasoning behaviors, using entropy at inference time for **cross-model routing** rather than truncating a single model’s trajectory.
>
> [1] Efficiently Scaling LLM Reasoning with Certaindex, 2024.
>
> **Q2.** Running with batch = 1 may exaggerate latency gains, and the authors should justify this design choice.
>
> **A2.** Please refer to Q2 in general response.
>
> **Q3.** Since the paper claims general acceleration of chain-of-thought reasoning, results across other reasoning domains (e.g., code, logical reasoning, or QA) would make the findings more convincing.
>
> **A3.** Please refer to Q4 in general response to see the results on addtional benchmarks.
>
> **Q4.** The paper ignores the memory overhead of maintaining both the LLM and SLM simultaneously. This dual-model setup increases GPU memory consumption and can reduce the number of requests processed in parallel. It’s possible that this constraint is why the experiments use batch size = 1, which should have been discussed explicitly.
>
> **A4.** Please refer to Q3 in general response.
>
> **Q5.** The method is not compared to alternative reasoning optimization approaches such as early termination or compression-based techniques, which also shorten reasoning traces. A fair evaluation should include or discuss these baselines to clarify the relative contribution.
>
> **A5.** We have included in initial submission a comparison between R-Stitch and the training-free early-exit method DEER(appendix A.7), showing that **the two approaches are complementary**.
> 1. DEER shortens the reasoning trajectory by stopping generation once confidence is high, while R-Stitch reduces per-token latency and removes substantially more tokens by invoking the LLM only when needed and leveraging concise SLM reasoning chains.
> 2. On AIME, token count drops 11739 → 9229 with DEER (−21%) and further to 4425 with R-Stitch (+DEER) (−62% total). On GPQA-D, token count decreases 10135 → 7084 with DEER (−30%) and to 1048 with R-Stitch (+DEER) (−90% total). Latency improves 171.76 s → 117.50 s → 18.15 s, and accuracy improves 23.74% → 27.78% → 30.81%, **achieving a 9.5× speedup over vanilla 7B decoding**.
>
> **Q6.** The approach’s robustness to entropy misprediction is unclear. If entropy fails to reflect uncertainty accurately (e.g., when the SLM is overconfident in incorrect outputs), the routing may degrade accuracy or fail to switch appropriately.
>
> **A6.** Below we clarify why entropy remains an effective routing signal in our method.
> 1. Section 3.2.1 shows that entropy reflects token-level uncertainty that correlates with reasoning errors, and Figure 4 further shows that it is statistically more effective than random switching under the same latency budget, even though it is not perfectly accurate on every token.
> 3. R-Stitch+ further improves robustness by replacing the fixed threshold with an RL-based router that learns a latency-aware policy and produces more consistent switching decisions.
>
> **Q7.** The computational overhead of computing token entropy is not thoroughly analyzed. Although entropy can be obtained from logits, its per-token computation and switching logic could add nontrivial latency, which should be measured or discussed.
>
> **A7.** Below we clarify the computational overhead of entropy-based routing. The extended details have been added to Appendix A.14 in the revised manuscript.
> 1. **The cost of computing token-level entropy is negligible.** Entropy is obtained directly from the LLM and SLM logits and adds only **0.028 ms/token** in our implementation. In the 7B–1.5B AIME setup (τ = 0.02), decoding **4530.60 tokens** takes **62.03 s**, while entropy evaluation contributes just **127 ms (≈0.2%)**, indicating negligible runtime burden.
> 2. **All switching-related overhead is already included in our latency measurements.** R-Stitch employs **partial prefill** to reuse cached representations and eliminate redundant computation during switching. The reported results reflect the **full end-to-end pipeline**, ensuring that the measured speedups remain representative of real deployment conditions.

---

### Official Review · Reviewer_mk62 · 2025-11-01

**Soundness:** 2
**Presentation:** 2
**Contribution:** 3
**Rating:** 4
**Confidence:** 4

**Summary:**

This paper focuses on efficient generation for long-CoT LLM. Long-CoT LLM normally generated extensively long trajectory, which is very time consumed. Speculative decoding (SpecDec) serves as traditional solution to speed it up. I.e. only use a LLM to generate when the tokens generated from the SLM are rejected. However, SpecDec requires a strict distribution alignment between LLM and SLM, and might falsely reject tokens that are correct but not aligned with LLM.

In this paper, the authors firstly observe that high entropy normally leave to incorrect generation, thus propose to use entropy itself as a hint for token rejection and acceptance, i.e. R-Stitch. If the entropy of a token from SLM is too high, the decoding is switched to LLM until meeting a new token with a low entropy, then the decoding is switched again to SLM. The authors further proposes another variant, R-Stich$^+$, that applies RL to train the model for better accuracy and efficiency gain.

Through extensive experiments, R-Stich and R-Stich$^+$ show comparable or better accuracy to the LLM and SpecDec, while significantly improving the speedup.

**Strengths:**

1. The observation of high entropy leading to incorrect trace is interesting, and well investigated. And the proposed method is weel aligned with the observation.
2. The experiments are thorough, with multiple LLMs and benchmarks, showing the benefits from R-Stitch.
3. The ablation study is well-designed, justifying the design choice.

**Weaknesses:**

1. The choice of SLM is not reasonable. L1-1.5B-Short is used as SLM, while the target model is DeepSeek-R1 family. As we know, SpecDec is efficient when both draft and target model's distribution is aligned. From Table 1, SpecDec's speedup is even worse than the target model alone, which is unreasonable. It's suggested to include new results with SLM from the same family.
2. Lack of baselines. Only two baselines are included here, LLM and SpecDec. It's suggested to include recent strong baselines for better justification, like:

[1] Reward-Guided Speculative Decoding for Efficient LLM Reasoning

[2] AdaEDL: Early Draft Stopping for Speculative Decoding of Large Language Models via an Entropy-based Lower Bound on Token Acceptance Probability

**Questions:**

### Suggestions
1. Better to highlight the best results in Table 1.

---

> ### Author Response · Authors · 2025-11-21
> **Response to Reviewer mk62**
>
> **Q1.** It's suggested to include new results with SLM from the same family.
>
> **A1.** Please refer to Q1 in general response.
>
> **Q2.** Lack of baselines. Only two baselines are included here, LLM and SpecDec. It's suggested to include recent strong baselines for better justification
>
> **A2.** We thank for pointing out the need for additional baselines.
> 1. To clarify, we **have compared R-Stitch with Eagle-3 in Appendix A.6** in the initial submission, a strong and widely adopted speculative decoding baseline integrated into vLLM. Across all reasoning benchmarks, **R-Stitch achieves consistently larger end-to-end acceleration, despite Eagle-3 using a heavily trained and smaller draft model**.
> 2. We show the comparisons with Reward-Guided Speculative Decoding (RSD) below, whose step-level switching depends on an external PRM. Because PRMs are trained for specific models and generalize poorly across different target models, **RSD’s routing decisions become unstable when the draft–target pair changes**. In contrast, R-Stitch performs fine-grained token-level routing without external models, achieving faster inference with comparable accuracy. All numbers report accuracy (%) and latency (s) under 8k decoding budgets, **using the same model pair as in Table 1 of the main paper**. Extended results have been added to Appendix A.15 in the revised manuscript.
> 3. For AdaEDL, direct experimental comparison is not possible because it has not been open-sourced. Conceptually, AdaEDL still depends on high distributional consistency between the draft and target models and only reduces the verification horizon. As a result, **it cannot leverage community SLMs that are concise but distributionally inconsistent**, which is precisely the scenario that R-Stitch is designed to address.
>
> | Method       |     AIME     |     AMC      |    MATH     |   Minerva   |
> |:------------ |:------------:|:------------:|:-----------:|:-----------:|
> | LLM-7B       | 33.33/86.63  | 66.27/63.15  | 86.00/38.34 | 31.62/41.79 |
> | RSD-7B       | 30.00/69.73  | 66.27/54.12  | 81.60/25.12 | 33.80/54.37 |
> | R-Stitch-7B  | 40.00/62.03  | 69.88/34.89  | 87.00/16.61 | 33.09/18.98 |

---

### Official Review · Reviewer_whxy · 2025-11-01

**Soundness:** 2
**Presentation:** 3
**Contribution:** 2
**Rating:** 6
**Confidence:** 2

**Summary:**

Authors propose entropy-guided hybrid decoding which switches between a SLM and LLM during chain-of-thought reasoning:

1. They uses normalized token entropy H_t = -Σp_{t,i}log(p_{t,i})/log(V) as an uncertainty proxy
2. SLM to LLM when entropy > τ (high uncertainty)
   LLM to SLM when entropy <= τ (low uncertainty, save computation)
3.  Maintains separate caches with partial prefill to minimize switching overhead

They add a learned routing policy via RL:
- Lightweight router decides when to invoke LLM for high-entropy tokens
- R = r_acc - λ·r_acc·L (penalizes latency only when correct)
- Linear regression models T(N_inf, N_kv) to avoid profiling overhead during training
- Group-normalized advantages for policy gradient

## Here are my thoughts
The paper conflates two distinct phenomena:
- Model disagreement (what speculative decoding addresses)
- Verbosity differences (SLM produces shorter traces)

The core claim that "speculative decoding's rigid token-level consistency prevents using SLM's conciseness" is misleading?? Speculative decoding doesn't prevent conciseness, it enforces correctness. If the SLM produces a correct but shorter solution, speculative decoding will accept it. The issue is that SLMs are often wrong, not just verbose.

2. Unfair Experimental Comparisons?
- Speculative decoding baseline uses high-agreement pairs (Distill-7B + Distill-1.5B) while R-Stitch uses low-agreement pairs (LLM + L1-Short)
- Is this maube backwards? Speculative decoding should get the favorable pairing?
- Are these cherry-pick scenarios where speculative decoding fails while giving R-Stitch optimal conditions?

3. Lack of Correctness Guarantees:
- R-Stitch discards SLM tokens and overwrites with LLM when entropy is high
- This creates a correctness risk: What if the SLM was actually right but uncertain? The LLM might introduce errors
- It would be nice to have an analysis of cases where switching to LLM hurts accuracy
- Table 1 shows accuracy drops in many settings (e.g., 7B on AIME: 33.33->30.00 at τ=0.03)

4. Entropy Is a Weak Proxy?
The empirical analysis (Section 3.2.1) is maybe superficial?
- Figure 3a: "Incorrect answers have higher entropy" - but correlation does not equal causation
- No comparison to other uncertainty measures (variance, top-k probability gaps, etc.)
- Figure 3c: "Harmful tokens have higher preceding entropy" - but the effect size is tiny (~0.028 vs ~0.024)
- 10.65% of tokens exceed entropy 0.1 - does this means the routing decision is rarely invoked?

5. Speedup Claims
- Peak speedups (3.00x, 3.85x, 4.10x) come with significant accuracy drops
- At τ=0.02 (claimed "sweet spot"), accuracy often decreases 2-5 points
- Speculative decoding maintains accuracy by construction
- It would be nice to report Pareto frontiers


7. Method
- Entropy-based routing is already used in early exit, mixture-of-experts, etc.
- Is the "stitching" framing marketing? This is just conditional execution?
- R-Stitch+ is standard REINFORCE with a domain-specific reward
- The related work section (A.5) acknowledges EAGLE, Griffin, Hydra do similar things but claims they "increase consistency" while R-Stitch "exploits inconsistency" - I might be wrong, but this feels like  a false dichotomy

9. Minor Issues
- No error bars or confidence intervals (use SE if you can)
- Small test sets (30 samples on AIME)
- Table 1 shows accuracy increasing from 66.27 (LLM) to 77.11 (R-Stitch τ=0.001) on AMC 8k - this suggests variance, not real improvement?

## Fundamental Question

Why not just use the LLM with early stopping? If the issue is that LLMs are verbose, methods like DEER (which the paper briefly mentions) achieve similar latency reductions without the complexity of dual models. The paper doesn't convincingly argue why heterogeneous model collaboration is necessary?


I'm recommending weak accept, but I can move my score up if you address my issues. Thanks.

**Strengths:**

Strengths are in the above review.

**Weaknesses:**

Weaknesses are in the above review.

**Questions:**

Why not just use the LLM with early stopping? If the issue is that LLMs are verbose, methods like DEER (which the paper briefly mentions) achieve similar latency reductions without the complexity of dual models. The paper doesn't convincingly argue why heterogeneous model collaboration is necessary?

---

> ### Author Response · Authors · 2025-11-21
> **Response to Reviewer whxy(Part 1)**
>
> **Q1.** Speculative decoding doesn't prevent conciseness, it enforces correctness. If the SLM produces a correct but shorter solution, speculative decoding will accept it. The issue is that SLMs are often wrong, not just verbose.
>
> **A1.** Speculative decoding accepts SLM tokens only when they match the LLM’s next-token distribution, and **this strict token-level consistency is the core limitation we highlight**.
>
> 1. Under this rule, **concise or semantically correct SLM continuations are frequently rejected** whenever their tokenization diverges from the LLM, preventing speculative decoding from benefiting from SLMs that produce shorter reasoning traces.
> 2. This limitation becomes significant in today’s open-source ecosystem, where **many SLMs generate much more compact reasoning but have low distributional consistency** with strong LLM families such as Qwen or DeepSeek-R1.
> 3. **R-Stitch overcomes this gap by switching dynamically rather than enforcing alignment**, enabling the system to exploit the conciseness of community SLMs while preserving the reliability of the LLM.
>
>
> **Q2.** Speculative decoding baseline uses high-agreement pairs (Distill-7B + Distill-1.5B) while R-Stitch uses low-agreement pairs (LLM + L1-Short) Is this maube backwards? Speculative decoding should get the favorable pairing? Are these cherry-pick scenarios where speculative decoding fails while giving R-Stitch optimal conditions?
>
> **A2.** Please refer to Q1 in general response.
>
> **Q3.** R-Stitch discards SLM tokens and overwrites with LLM when entropy is high. This creates a correctness risk: What if the SLM was actually right but uncertain? The LLM might introduce errors. Entropy Is a Weak Proxy? The empirical analysis (Section 3.2.1) is maybe superficial?
>
> **A3.** Below we clarify why entropy remains an effective routing signal in our method.
> 1. Section 3.2.1 shows that entropy captures token-level uncertainty that correlates with reasoning errors. Figure 4 further demonstrates that, while entropy is not perfectly accurate on every token, it is statistically more effective than random switching.
> 2. R-Stitch+ further improves robustness by replacing the fixed threshold with an RL-based router that learns a policy and yields more stable switching decisions.
>
> **Q4.** Is the "stitching" framing marketing? This is just conditional execution?
>
> **A4.** The “stitching” terminology is not marketing but follows an established line of work on model stitching and cross-model composition, as noted in our related work(line 151-161). Prior studies already use this framing for combining models or intermediate representations. Our contribution is to adapt this paradigm to reasoning-time inference, showing that **token-level, bidirectional stitching enables flexible collaboration between SLMs and LLMs**.
>
> **Q5.** Entropy-based routing is already used in early exit, mixture-of-experts, etc.
>
> **A5.** Below we clarify how our use of entropy differs.
> 1. Early-exit methods[1,2] use entropy to stop a single model’s decoding when confidence becomes high. R-Stitch addresses a different problem: rather than truncating one model’s trajectory, it uses entropy to coordinate two models with **complementary reasoning characteristics**.
> 2. MoE routing[3] incorporates entropy during training to stabilize expert selection. R-Stitch instead uses entropy at inference time as a routing signal between the SLM and LLM, turning uncertainty directly into an efficiency gain.
>
> [1] Efficiently Scaling LLM Reasoning with Certaindex, 2024.
>
> [2] Dynamic Early Exit in Reasoning Models, 2024.
>
> [3] Improving Routing in Sparse Mixture of Experts with Graph of Tokens, 2023.
>
>
>
>
> **Q6.** The related work section (A.5) acknowledges EAGLE, Griffin, Hydra do similar things but claims they "increase consistency" while R-Stitch "exploits inconsistency".
>
> **A6.** These prior methods require substantial training to **increase SLM–LLM consistency**, pushing the SLM to behave more like the LLM. Our setting is fundamentally different: we observe that some open-source SLMs produce **much more concise reasoning** despite having **low distributional alignment** with LLMs. Instead of training an SLM to mimic the LLM, which would also make it verbose, R-Stitch directly leverages these concise but low-consistency SLMs through dynamic switching.

---

> > ### Author Response · Authors · 2025-11-21
> > **Response to Reviewer whxy(Part 2)**
> >
> > **Q7.** Table 1 shows accuracy increasing from 66.27 (LLM) to 77.11 (R-Stitch τ=0.001) on AMC 8k - this suggests variance, not real improvement?
> >
> > **A7.** The higher accuracy on AMC 8k is not variance but a consequence of how RL-trained reasoning LLMs behave under a strict 8k budget. These models often produce long, verbose chains of thought, causing the vanilla LLM to exceed the limit and get truncated, which harms accuracy. R-Stitch alleviates this by allowing the concise SLM to handle low-uncertainty spans, **enabling the LLM to complete the reasoning within a much shorter trajectory**. The reduced chain length avoids truncation and lowers KV-cache memory usage, supporting the advantage of R-Stitch in both accuracy under length constraints and memory efficiency. The explaination has been added to experiment section (line 397) in the revised manuscript.
> >
> > **Q8.** Why not just use the LLM with early stopping? If the issue is that LLMs are verbose, methods like DEER (which the paper briefly mentions) achieve similar latency reductions without the complexity of dual models. The paper doesn't convincingly argue why heterogeneous model collaboration is necessary?
> >
> > **A8.** We have included in the initial submission a comparison between R-Stitch and the training-free early-exit method **DEER** (Appendix A.7), showing that the two approaches are **complementary rather than interchangeable**.
> > 1. DEER shortens the reasoning trajectory by stopping generation once confidence is high, while R-Stitch reduces per-token latency and removes substantially more tokens by invoking the LLM only when needed and leveraging concise SLM reasoning chains.
> > 2. On AIME, token count drops 11739 → 9229 with DEER (−21%) and further to 4425 with R-Stitch (+DEER) (−62% total). On GPQA-D, token count decreases 10135 → 7084 with DEER (−30%) and to 1048 with R-Stitch (+DEER) (−90% total). Latency improves 171.76 s → 117.50 s → 18.15 s, and accuracy improves 23.74% → 27.78% → 30.81%, **achieving a 9.5× speedup over vanilla 7B decoding**.

---

### Official Review · Reviewer_7aFp · 2025-11-02

**Soundness:** 2
**Presentation:** 3
**Contribution:** 2
**Rating:** 4
**Confidence:** 3

**Summary:**

The paper proposes R-Stitch, a bidirectional, entropy-guided collaboration between a small language model (SLM) and a large language model (LLM) for chain-of-thought reasoning. Decoding starts on the SLM, switches to the LLM when the SLM’s token entropy exceeds a threshold, and switches back once the LLM becomes confident, which preserves the SLM’s concise spans while using the LLM for hard tokens. To keep switching efficient, the system maintains separate KV caches and performs partial prefill so each model only recomputes tokens generated since the last switch. The R-Stitch+ variant adds a lightweight router trained with a latency-aware reward that penalizes runtime only when the answer is correct and relies on a profiled latency estimator to avoid online timing. Implemented in vLLM and evaluated on five math benchmarks with 7B, 14B, and 32B models, the method reduces per-sample wall-clock latency on a single A100 at batch size one, delivering roughly 1.4×–3.0× speedups at 7B/14B and up to around 4× at 32B under 8k–16k budgets while maintaining accuracy close to full LLM decoding.

**Strengths:**

* The core algorithm is simple and training free in its base form, using a clear entropy threshold to switch between SLM and LLM in both directions so that the system exploits concise SLM spans without sacrificing reliability on high-uncertainty tokens.
* The method is well motivated by an empirical analysis showing that incorrect answers have higher token entropy and that most tokens are very low entropy, which justifies entropy as a routing signal.
* The systems design is thoughtful, with explicit KV-cache management and partial prefill that reuses past caches on each model to avoid redundant attention and reduce switching overhead.

**Weaknesses:**

* The router is only described as a “lightweight” module fed by hidden states; its architecture, parameter count, placement, and per-token overhead are not reported, so deployability and reproduction costs are unclear.
* All latency results use a single GPU with batch size one. The current implementation only supports batch size one because switching happens at the token level. Real-world throughput under concurrent traffic is unknown.
* The system runs two engines with separate KV caches. This increases VRAM usage and system complexity. Partial prefill reuse is described, but the paper does not quantify memory costs or switching overhead.
* Performance depends on the entropy threshold. The paper tunes the threshold by sweeping values across a grid. There is no automatic rule that transfers across datasets and model pairs, which raises generability concerns of real-world deployments.
* The overall router design will make the system hard to scale and deploy comparing with methods like Eagle-3 etc.

**Questions:**

* Please specify the router: architecture, parameter count, compute placement, input features, and measured per-token overhead (and its share of end-to-end latency).
* For R-Stitch+, how robust is the learned policy across domains and model sizes? Please include cross-domain transfer results.

---

> ### Author Response · Authors · 2025-11-21
> **Response to Reviewer 7aFp**
>
> **Q1.** The router is only described as a “lightweight” module fed by hidden states; its architecture, parameter count, placement, and per-token overhead are not reported, so deployability and reproduction costs are unclear.
>
> **A1.** The router is explicitly designed to be lightweight, and **its computational and memory overhead is negligible**. The extended details have been added to Appendix A.13 in the revised manuscript.
> 1. It is a small feed-forward network operating on SLM hidden states, composed of 6 FFN blocks with GELU and residual connections, followed by normalization and a two-head linear layer. The input matches the SLM hidden size (3584 for the 1.5B model), and **the total parameter count is only 0.17M**.
> 2. On an NVIDIA A100, the router adds **1.05 ms per invocation**. At τ = 0.02 on AIME with the 7B pair, it is invoked 350.38 times on average, giving **368 ms total overhead**.
> 3. The full inference time for the same setup is 62.03 s, so **the router contributes under 0.6% of end-to-end latency**, demonstrating that its deployability cost is minimal.
>
>
> **Q2.** All latency results use a single GPU with batch size one. The current implementation only supports batch size one because switching happens at the token level. Real-world throughput under concurrent traffic is unknown.
>
> **A2.** Please refer to Q2 in general response.
>
> **Q3.** The system runs two engines with separate KV caches. This increases VRAM usage and system complexity. Partial prefill reuse is described, but the paper does not quantify memory costs or switching overhead.
>
> **A3.** Please refer to Q3 in general response.
>
> **Q4.** The paper tunes the threshold by sweeping values across a grid. There is no automatic rule that transfers across datasets and model pairs, which raises generability concerns of real-world deployments.
>
> **A4.** Below we clarify the generality of the threshold and its applicability in real-world use.
> 1. The entropy threshold serves as a training-free routing signal. As shown in Section 3.2.1 and reinforced by our additional QA evaluations (Q4 in the general response), **the same threshold used for math tasks maintains stable performance across datasets and model pairs**, indicating strong cross-domain generality.
> 2. To further improve adaptivity, R-Stitch+ replaces the fixed threshold with a lightweight RL-based router, which learns latency-aware switching policies and mitigates occasional mispredictions. **This provides a more robust mechanism for deployment**.
>
>
> **Q5.** The overall router design will make the system hard to scale and deploy comparing with methods like Eagle-3 etc.
>
> **A5.** We **have already integrated R-Stitch into the vLLM serving framework** in the initial submission (mentioned in Implementation Details, line 362), and all experiments are conducted on top of this integration.
>
> 1. The integration enables efficient token-level model switching via vLLM’s scheduler with some backend modifications, showing that **the design is practical and compatible with existing large-scale inference systems**.
> 2. As shown in Appendix A.6, we compare R-Stitch with Eagle-3, a state-of-the-art speculative decoding method also deployed in vLLM. Under the same environment, R-Stitch delivers consistently higher end-to-end acceleration across all benchmarks, **demonstrating both its deployability and superior practical efficiency**.
>
> **Q6.** For R-Stitch+, how robust is the learned policy across domains and model sizes? Please include cross-domain transfer results.
>
> **A6.** Please refer to Q4 in general response.

---

> > ### Comment · Reviewer_7aFp · 2025-11-28
> >
> > Thanks the author for the detailed response. However, I still feel the scheduling part is non-trivial and needs some further clarification.
> >
> > Current inference engine usually observes the scheduling at each step could bottleneck the whole inference procedure, so they made efforts to make the scheduling part async. I am not sure how R-stitch fits into such design scheme and whether the overhead mentioned is fairly calculated.
> >
> > Additionally, the colocated of SLM and LLM could still make the KV Cache space complicated. Current inference engine usually requires to preserve the KV Cache on GPU. If SLM and LLM are colocated, the management of the KV Cache can be quite complicated in large-batch size inference scenarios. How the author address such issue?
> >
> > Lastly, the maximum scale of the experiment is done on 32B model. I am wondering the performance on really state-of-the-art open source models like Qwen 235B model, and DeepSeek-R1 model. The accuracy number on these frontier models would offer more confidence on the methods' performance.

---

> ### Author Response · Authors · 2025-12-01
> **Response to Reviewer 7aFp**
>
> Thank you for the detailed response. Below we provide further clarification to the follow-up questions:
>
> **Q7.** Current inference engine usually observes the scheduling at each step could bottleneck the whole inference procedure, so they made efforts to make the scheduling part async. I am not sure how R-stitch fits into such design scheme and whether the overhead mentioned is fairly calculated.
>
> **A7.** R-Stitch requires only minimal modification to fit into an async scheduling design.
> 1. While the model is running, the async scheduler updates request queues, allocates KV blocks, and performs post-processing operations based on the model output, for example setting the next-step input token id. **R-Stitch only adds two checks in this post-processing stage: determining which model executes the next step and updating the token id used for partial prefill when a switch occurs.** Since the SLM and LLM share the same context and KV block indices, no changes to the KV allocator are required.
> 2. We evaluated this within vLLM’s async engine. On MATH with an 8k budget using the DeepSeek-R1–Distill-Qwen-7B pair, standard async decoding averages 38.36 seconds, whereas R-Stitch $(\tau = 0.02)$ achieves 16.34 seconds. **The additional post-processing introduced by R-Stitch costs only 0.0269 seconds on average, which is about 0.16% of the total latency.**
>
> **Q8.** Additionally, the colocated of SLM and LLM could still make the KV Cache space complicated. Current inference engine usually requires to preserve the KV Cache on GPU. If SLM and LLM are colocated, the management of the KV Cache can be quite complicated in large-batch size inference scenarios. How the author address such issue?
>
> **A8.** Our implementation is built directly on vLLM, where both the SLM and LLM maintain their KV caches on GPU under the standard vLLM memory management.
> 1. R-Stitch maintains two KV caches, which is inherent to all speculative-decoding–style methods. Since the SLM is much smaller, its additional memory footprint remains small, as quantified in Q3 of the general response.
> 2. Because the two models share the same context, their token positions map to identical block IDs in vLLM, so no changes to the KV-block allocation logic are required. The scheduler only needs to determine which model executes the next step and whether partial prefill is needed, resulting in minimal modifications to the vLLM codebase.
>
> **Q9.** Lastly, the maximum scale of the experiment is done on 32B model. I am wondering the performance on really state-of-the-art open source models like Qwen 235B model, and DeepSeek-R1 model. The accuracy number on these frontier models would offer more confidence on the methods' performance.
>
> **A9.** We have conducted experiments up to 32B models under our available hardware resources. While we are unable to scale further to 200B-level models such as Qwen-2.5-235B or DeepSeek-R1 due to compute limitations, the results from 7B to 32B consistently demonstrate that R-Stitch provides stable speedup and maintains accuracy across model scales.

---

### Author Response · Authors · 2025-11-21
**Response to All Reviewers(Part 1)**

We sincerely thank all reviewers for their valuable and constructive feedback.
All reviewers acknowledge the importance of reasoning-time efficiency and recognize the practicality of our entropy-guided hybrid decoding approach.

> “The core algorithm is simple and training-free, using a clear entropy threshold to switch between SLM and LLM in both directions.” (R7aFp)
> “The observation that high-entropy tokens lead to incorrect traces is interesting and well investigated.” (Rmk62)
> “The use of entropy as a routing signal is well-motivated and empirically justified.” (RsaCs)
> “The systems design is thoughtful, with explicit KV-cache management and partial prefill that reuses past caches to reduce switching overhead.” (R7aFp)

These comments collectively indicate that **R-Stitch is viewed as a simple and effective framework** for efficient reasoning-time inference.


## **General Response**

**Q1.** Choice of model pairs and fairness of comparison (Reviewers whxy, mk62)

**A1.** Below we report results where speculative decoding uses both the model pairs in the manuscript and the same-family DeepSeek-R1–Distill-Qwen pairs with higher draft–target consistency (marked with *). All numbers are accuracy (%) and latency (s) under an 8k budget. We highlight in **bold** the best acceleration achieved without any accuracy drop. The extended comparisons have been added to Appendix A.9 in the revised manuscript.

| Method        |     AIME     |     AMC      |    MATH     |   Minerva   |
|:------------- |:------------:|:------------:|:-----------:|:-----------:|
| LLM 7B        | 33.33/86.63  | 66.27/63.15  | 86.00/38.34 | 31.62/41.79 |
| SpecDec 7B    | 36.67/201.23 | 69.88/95.42  | 87.00/48.71 | 34.19/56.59 |
| R-Stitch 7B   | **40.00/62.03**  | **69.88/34.89**  | **87.00/16.61** | **33.09/18.98** |
| SpecDec* 7B   | 33.33/68.65 | 71.08/58.53  | 89.20/36.86 | 34.19/43.33 |
| R-Stitch* 7B  | 36.67/69.73 | 68.67/61.16  | 89.20/40.55 | 36.76/41.15      |
| LLM 14B       | 43.33/153.20 | 68.67/101.76 | 86.00/41.66 | 35.29/48.02 |
| SpecDec 14B   | 50.00/139.10 | 68.67/95.59  | 82.80/45.88 | 34.93/53.02 |
| R-Stitch 14B  | **43.33/87.61**  | **69.88/41.82**  | **85.20/20.53** | **35.66/22.16** |
| SpecDec* 14B  | 50.00/127.59 | 74.70/71.39  | 83.80/37.02 | 33.46/40.95 |
| R-Stitch* 14B | 43.33/112.23 | 74.70/73.24  | 85.20/35.41 | 35.66/38.89      |

1. Speculative decoding relies on high draft–target consistency for acceleration, and even same-family pairs may still exhibit insufficient consistency to yield strong speedups. This is reflected in the table: when consistency is higher (the * pairs), speculative decoding improves but remains limited.
2. Because speculative decoding fundamentally depends on strict token-level agreement, it cannot leverage community SLMs that are concise but distributionally inconsistent with the LLM. In contrast, R-Stitch removes this constraint and can exploit such concise SLMs to shorten the reasoning trajectory and achieve larger acceleration gains that speculative decoding cannot obtain.
3. Open-sourced reasoning LLMs such as Qwen tend to produce verbose chains, and same-family SLMs inherit this verbosity, limiting acceleration. R-Stitch can use concise SLMs even with lower consistency to reduce reasoning length.

**Q2.** Batch size and evaluation realism (Reviewers 7aFp, saCs)

**A2.** We clarify the effect of batch size on R-Stitch performance. The extended discussions have been added to Appendix A.10 in the revised manuscript.
1. Since R-Stitch dynamically switches between the SLM and LLM at the token level, its acceleration ratio naturally decreases as batch size increases. **This trend is common to all speculative decoding–style methods, where alternating model invocations reduce GPU utilization as requests must synchronize at switching steps.**
2. We evaluate R-Stitch and speculative decoding on the MATH dataset using DeepSeek-R1-Distill-Qwen-7B within vLLM, with τ = 0.02. The full LLM achieves 86.00% accuracy, while both acceleration methods reach 87.00%, showing no accuracy degradation. Latency values represent average per-sample latency after completing the dataset under each batch size. As shown below, latency increases with larger batches and the benefit of acceleration diminishes, but **R-Stitch consistently achieves lower latency than speculative decoding** across all settings. Its entropy-guided, bidirectional routing better leverages the concise reasoning behavior of small models, while speculative decoding’s strict consistency constraint limits its efficiency.

| **Method / Batch** | **1** | **2** | **4** | **8** | **16** | **32** | **64** | **128** |
|:-------------------:|:----:|:----:|:----:|:----:|:----:|:----:|:----:|:----:|
| **LLM** | 38.34 | 31.58 | 22.07 | 16.11 | 11.08 | 8.42 | 6.52 | 5.97 |
| **SpecDec** | 36.86 | 31.61 | 24.29 | 22.04 | 19.20 | 17.88 | 15.32 | 14.01 |
| **R-Stitch** | 16.61 | 20.48 | 17.98 | 14.98 | 12.02 | 10.61 | 7.43 | 6.55 |

---

> ### Author Response · Authors · 2025-11-21
> **Response to All Reviewers(Part 2)**
>
> **Q3.** Dual KV caches and memory overhead (Reviewers 7aFp, saCs)
>
> **A3.** We acknowledge that **R-Stitch, similar to all speculative decoding–style methods, must maintain separate KV caches for the SLM and LLM**. To quantify the actual overhead, we measured model weights and KV cache footprints in **vLLM** under BF16 precision. The extended discussions have been added to Appendix A.11 in the revised manuscript.
>
> | **Model size** | **Per-token KV (MB)** | **Model weights (MB)** | **Total @8k (MB)** | **Total @16k (MB)** |
> |----------------|-----------------------|------------------------|--------------------|---------------------|
> | 1.5B | 0.028 | 3036 | 3265 | 3494 |
> | 7B | 0.055 | 15640 | 16091 | 16542 |
> | 14B | 0.189 | 29275 | 30824 | 32374 |
> | 32B | 0.253 | 63890 | 65958 | 68026 |
>
> When paired with a 7B model under 16k context, the 1.5B SLM adds only 17.44% to total memory usage, and this ratio further drops to 9.74% for a 14B model and 4.89% for a 32B model. Overall, **the additional memory cost from enabling R-Stitch remains marginal compared with LLM**, while yielding significant end-to-end latency reductions.
>
> **Q4.** Evaluation scope and cross-domain generalization (Reviewers 7aFp, saCs)
>
> **A4.** We evaluate R-Stitch($\tau=0.02$) and R-Stitch+ beyond mathematical reasoning on four QA and reasoning benchmarks (GPQA-D[1], ZebraLogicBench[2], CRUXEval[3], and MMLU-Redux[4]), reporting Accuracy (↑) and Latency (↓). For R-Stitch+, the router is trained only on math datasets and directly applied to these new domains without any retraining. The extended content have been added to Appendix A.12 in the revised manuscript.
>
> | Method | GPQA-D | ZebraLogicBench | CRUXEval | MMLU-Redux |
> |:--|:--:|:--:|:--:|:--:|
> | LLM-7B | 23.74/171.76 | 21.09/91.85 | 70.25/14.54 | 74.64/20.10 |
> | SpeDec | 24.24/152.23 | 20.22/72.03 | 70.50/14.78 | 74.04/19.37 |
> | R-Stitch | 35.86/44.57 | 20.78/48.23 | 69.63/8.25 | 74.04/8.12 |
> | R-Stitch+ | 34.34/39.02 | 20.82/45.12 | 69.88/9.15 | 74.11/8.52 |
>
> 1. R-Stitch maintains accuracy comparable to LLM across these datasets while delivering clear end-to-end speedups.
> 2. R-Stitch+ stably provides slightly better accuracy using a router trained only on math data, indicating that **the learned policy transfers reasonably well across different tasks** without retraining.
>
> [1] GPQA: A Graduate-Level Google-Proof Q&A Benchmark
>
> [2] ZebraLogic: On the Scaling Limits of LLMs for Logical Reasoning
>
> [3] CRUXEval: A Benchmark for Code Reasoning, Understanding and Execution
>
> [4] Are We Done with MMLU?

---

### Meta-Review · Area_Chair_S6SX · 2026-01-07

**Summary:**

Long chain-of-thought reasoning makes inference expensive, and existing acceleration methods (notably speculative decoding) can under-deliver when the draft–target agreement is low. The authors propose a token-level, entropy-guided switching between an SLM and an LLM with KV-cache reuse (“stitching”), and several reviewers find the systems implementation thoughtful (vLLM integration, partial prefill, cache reuse).

However, the main critical concerns are: (i) novelty relative to speculative decoding / early-exit / conditional compute, with some reviewers viewing the method as incremental; (ii) fairness and clarity of comparisons (choice of model pairs, baselines, and whether speedups can only be realized with a batch size of 1); and (iii) memory overhead from dual KV caches. Additional concerns include robustness/generalization of the entropy threshold (or routing policy).

After the rebuttal, the authors added additional experiments on more model pairs for both the proposed approach and speculative decoding, as well as results with larger batch sizes. Overall, I lean borderline reject of this paper because with the new rebuttal results, the good results of the method seem to necessarily come from a strong, short reasoning models compressed from long reasoning models (L1 here). When using a different pair (* entries in Q1 of general response), the latency reductions diminish significantly. Also, with larger batch sizes, the latency reductions also diminishe and underperforms the vanilla decoding baseline, which means the efficiency is hard to be realized in deployment.

**Reviewer Concerns:**

Concerns addressed by the rebuttal:

1. Router details and overhead for R-Stitch+
2. KV-cache memory overhead
3. Scheduling overhead concerns


Concerns that remain outstanding (or only partially addressed):
1. Fairness of speculative decoding comparisons / model-pair choice: The authors added comparisons where speculative decoding uses same-family pairs with higher draft–target consistency. The results show that under this setting the model's previous latency reductions diminish a lot, and it seems a small, short reasoning model that comes from a long reasoning model like L1 is necessary.
2. Batch size / throughput realism: While the original experiments emphasized batch size 1, the authors added a batch-size sweep (including larger batches) and discuss the expected trend that token-level switching reduces utilization as batch grows. While these results show R-Stitch still beat speculative decoding across settings, it starts to underperform vanilla decoding baselines when batch size grow to 16 or larger, which raises further doubt about realism of the speedup.

**Reviewer Scores:**

No reviewers explicitly mention review score change. From my perspective:

1. Reviewer 7aFp (4 → 4): Their main issues were missing router details, batch-size realism, and KV/memory overhead. After the rebuttal, the batch-size realism is still an issue.
2. Reviewer mk62 (4 → 4): They requested same-family SLM comparisons and more baselines (e.g., reward-guided speculative decoding). The rebuttal directly adds these, but they results show that the proposed approach is not performing as well as in the original submission in this case.
3. Reviewer whxy (6 → 6):  They were already slightly positive but raised conceptual objections about how speculative decoding relates to conciseness and asked for clearer tradeoffs and baselines (including early-exit comparisons). The rebuttal strengthens the justification and adds complementary evidence.
4. Reviewer saCs (2 → 2): Their rejection centered on limited novelty, unrealistic batch=1 setup, narrow evaluation scope, memory overhead, and missing overhead analysis. The rebuttal substantially addresses the latter four (batch-size sweep, cross-domain results, memory overhead, entropy/routing overhead), but novelty concerns likely remain.The batch-size sweep results may further confirm the reviewer's concern.

---

### Decision · Program_Chairs · 2026-01-26

Reject